# M³HF: Multi-agent Reinforcement Learning from Multi-phase Human Feedback of Mixed Quality

**Ziyan Wang** [1] [*]  **Zhicheng Zhang** [2]  **Fei Fang** [2]  **Yali Du** [1]

## Abstract

Designing effective reward functions in multi-agent reinforcement learning (MARL) is a significant challenge, often leading to suboptimal or misaligned behaviors in complex, coordinated environments. We introduce Multi-agent Reinforcement Learning from Multi-phase Human Feedback of Mixed Quality (M³HF), a novel framework that integrates multi-phase human feedback of mixed quality into the MARL training process. By involving humans with diverse expertise levels to provide iterative guidance, M³HF leverages both expert and non-expert feedback to continuously refine agents' policies. During training, we strategically pause agent learning for human evaluation, parse feedback using large language models to assign it appropriately and update reward functions through predefined templates and adaptive weights by using weight decay and performance-based adjustments. Our approach enables the integration of nuanced human insights across various levels of quality, enhancing the interpretability and robustness of multi-agent cooperation. Empirical results in challenging environments demonstrate that M³HF significantly outperforms state-of-the-art methods, effectively addressing the complexities of reward design in MARL and enabling broader human participation in the training process. Code is available at `https://github.com/cooperativex/M3HF`

## 1. Introduction

Designing effective reward functions for reinforcement learning (RL) agents is a well-known challenge, particularly in complex environments where the desired behaviors are intricate or the rewards are sparse (Singh et al., 2009; Ng et al., 2000). This difficulty is magnified in multi-agent reinforcement learning (MARL) settings, where agents must not only learn optimal individual behaviors but also coordinate with others, leading to an exponential increase in task complexity (Zhang et al., 2021; Oroojlooy & Hajinezhad, 2023; Du et al., 2023). Sparse or hard-to-learn rewards can severely hinder the learning process, causing agents to converge slowly or settle on suboptimal policies (Andrychowicz et al., 2017; Pathak et al., 2017). In such scenarios, relying solely on environmental rewards may be insufficient for effective learning. Incorporating human feedback has thus emerged as a promising approach (Christiano et al., 2017; Knox & Stone, 2009; Ho & Ermon, 2016), since human guidance can provide additional, informative signals that help agents navigate complex tasks more efficiently when intrinsic rewards are inadequate.

To leverage human expertise in accelerating the learning process of MARL agents, we propose the Multi-phase Human Feedback Markov Game (MHF-MG), an extension of the Markov Game that incorporates human feedback across multiple generations of learning. At each generation, agents gather experiences using their current policies but may still struggle under the original reward function. Humans then observe the agents' behaviors, offering feedback that reflects the discrepancy between their own (potentially more expert) policy and the agents' policies. Building on the MHF-MG, we develop Multi-agent Reinforcement Learning from Multi-phase Human Feedback of Mixed Quality (M³HF), which operationalizes the MHF-MG by directly integrating feedback into the agents' reward functions. This framework utilizes large language models (LLMs) to parse human feedback of various quality levels, employs predefined templates for structured reward shaping, and applies adaptive weight adjustments to accommodate mixed-quality signals.

In summary, we make the following contributions: (1) We propose the **MHF-MG** to address reward sparsity and com-

---
[*]Work done during a visit to CMU. [1]Department of Informatics, King's College London, London, United Kingdom [2]Software and Societal Systems Department, Carnegie Mellon University, Pittsburgh, PA, USA. Correspondence to: Ziyan Wang <ziyan.wang@kcl.ac.uk>.

*Proceedings of the 42ⁿᵈ International Conference on Machine Learning*, Vancouver, Canada. PMLR 267, 2025. Copyright 2025 by the author(s).

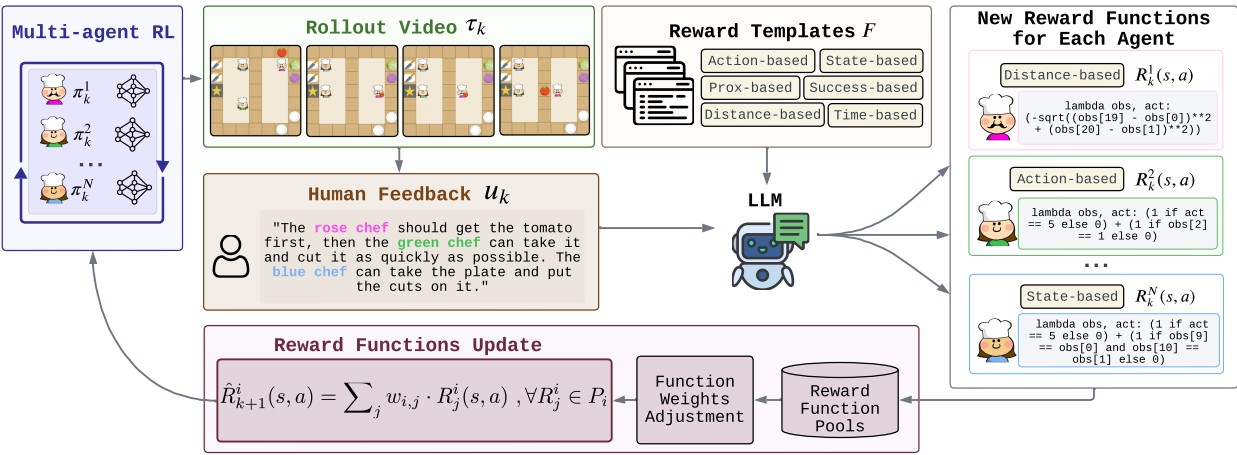

*Figure 1.* Workflow of the M³HF method. Each generation $k \in (0, .., K-1)$ begins with Multi-agent RL training. Agents generate rollout videos $\tau_k$ for human evaluation. Human feedback $u_k$ is parsed by a Large Language Model (LLM) into agent-specific instructions. The LLM then selects appropriate reward function templates $f \in F$ and parameterizes them based on the parsed feedback. New reward functions $R_k^i(s, a)$ are added to each agent's reward function pool $P_i$, with weights $w_{i,m}$ adjusted using performance-based criteria. The updated reward functions $\hat{R}_{k+1}^i(s, a)$ guide the next generation of agents' training, creating a loop of agents learning from feedback.

plexity in MARL through iterative human guidance; (2) We develop the **M³HF** framework, which leverages LLMs to parse diverse human feedback and dynamically incorporates it into agents' reward functions; (3) We provide a theoretical analysis justifying the use of rollout-based performance estimates and offering a weight decay mechanism that mitigates the impact of low-quality feedback. Extensive experiments in Overcooked demonstrate that M³HF consistently outperforms strong baselines, ultimately providing a robust and flexible method for enhancing multi-agent cooperation under challenging reward structures.

## 2. Related Work

**Multi-Agent Reinforcement Learning (MARL)** has been extensively studied to enable agents to learn coordinated behaviors in shared environments (Du et al., 2023; Yu et al., 2022). Traditional MARL approaches often rely on predefined reward functions and suffer from scalability and stability issues arising from the non-stationarity introduced by multiple learning agents (Busoniu et al., 2008; Canese et al., 2021). However, designing appropriate reward functions in MARL remains a significant challenge due to the complexity of agent interactions and the potential for conflicting objectives. To address this, researchers have explored various techniques for reward design. These include credit assignment methods (Nguyen et al., 2018; Zhou et al., 2020), reward shaping (Mannion et al., 2018), and the use of intrinsic rewards (Du et al., 2019). Furthermore, reward decomposition approaches have been proposed to balance individual and team objectives, such as separating rewards

into contributions from self and nearby agents (Zhang et al., 2020), or combining dense individual rewards with sparse team rewards (Wang et al., 2022).

**Reinforcement Learning from Human Feedback (RLHF)** has emerged as a promising avenue to address the limitations of handcrafted reward functions. Christiano et al. (2017) introduced methods for training agents using human preferences to shape the reward function, demonstrating that human feedback can significantly enhance policy learning. Building on this, Lee et al. (2021) proposed PEBBLE, leveraging unsupervised pre-training and experience relabeling to improve feedback efficiency in interactive RL settings.

While RLHF has been successfully applied to train Large Language Models (LLMs) (Ouyang et al., 2022; Shani et al., 2024; Wang et al., 2025; Hu et al., 2025; Liu et al., 2024), these approaches primarily focus on aligning LLM outputs with human preferences through single-turn interactions and scalar reward signals. Recently, PbMARL (Zhang et al., 2024) applied RLHF to MARL by utilizing offline pairwise preference comparisons extracted from pre-collected simulated policy data, but remained limited to an offline single-phase feedback scenario without dynamic, iterative human interactions. In contrast, our work incorporates multi-phase, mixed-quality human feedback directly into the reinforcement learning loop during online training in a multi-agent environment, enabling more flexible and scalable reward shaping.

**Language Models in Reward Design and Policy Learning.** Recent advancements in LLMs have created oppor-

tunities for incorporating natural language guidance into reinforcement learning (RL) (Liu et al., 2022; Chen et al., 2024). For instance, Ma et al. (2024) introduced EUREKA, a method employing code-generating LLMs to craft sophisticated reward functions at a human-expert level. However, extending methods such as EUREKA to multi-agent settings remains non-trivial, as independently performing rollouts to optimize rewards for each agent can become prohibitively expensive and computationally intensive. Similarly, Liang et al. (2023) proposed Code as Policies, where language model programs are used for embodied control, allowing agents to interpret and execute high-level instructions.

Other recent works have further leveraged LLMs to translate linguistic instructions into reward functions and policies. Yu et al. (2023) utilized language instructions to shape rewards for robotic skills, effectively aligning robot behaviors with human guidance. Kwon et al. (2023) highlighted the capacity of language models to capture nuanced human preferences during reward design. In human-AI collaboration (Zou et al., 2025), Hu & Sadigh (2023) demonstrated how language-instructed RL could foster improved coordination between humans and agents, Wang et al. (2024) uses LLMs to convert language constraints into cost signals for agents. Additionally, Liang et al. (2024) proposed leveraging language model predictive control to iteratively refine RL policy learning from human feedback more quickly.

Relatedly, Klissarov et al. (2023) presented MOTIF, adopting LLM-generated intrinsic rewards from textual task specifications to promote effective exploration in single-agent environments. Although MOTIF effectively leverages language grounding, it is inherently limited to single-agent contexts and does not incorporate iterative human interactions or complex multi-agent cooperation scenarios. In contrast, our approach specifically employs LLMs to parse multi-phase, mixed-quality human feedback dynamically to iteratively refine reward functions in complex multi-agent coordination problems.

**Multi-phase Human Feedback.** Prior works have considered the role of iterative and multi-phase human feedback in reinforcement learning. Yuan et al. (2022) and Sumers et al. (2022) explored multi-phase bidirectional interactions between humans and agents through predefined communication protocols, which, while structured, limit the flexibility of feedback. Zhi-Xuan et al. (2024) examined the use of human demonstrations via trajectories to convey intentions, requiring humans to perform the task themselves, which can be resource-intensive. Early attempts by Chen et al. (2021) and Zhang et al. (2023) delved into language for task generalization and policy explanation but were constrained to single-agent domains.

## 3. Preliminaries

We consider a **Markov Game** (Littman, 1994), defined by the tuple $\langle \mathcal{N}, \mathcal{S}, \mathcal{A}, P, R, \gamma \rangle$, in multi-agent reinforcement learning (MARL). Here, $\mathcal{N} = \{1, 2, \ldots, N\}$ represents the set of agents. The state space $\mathcal{S}$ encompasses all possible configurations of the environment, while the action space $\mathcal{A}$ denotes the set of actions available to each agent. At each time step $t$, the environment is in a state $s_t \in \mathcal{S}$. Each agent $i \in \mathcal{N}$ selects an action $a_t^i \in \mathcal{A}$ according to its policy $\pi^i(a_t^i|s_t)$. The joint action $\mathbf{a}_t = (a_t^1, a_t^2, \ldots, a_t^N)$ leads to a state transition to $s_{t+1}$ according to the transition function $P(s_{t+1}|s_t, \mathbf{a}_t)$. The agents receive a shared reward $r_t = R(s_t, \mathbf{a}_t)$, where $R : \mathcal{S} \times \mathcal{A}^N \to \mathbb{R}$ is the reward function, and $\gamma \in [0, 1)$ is the discount factor. The objective for each agent is to learn a policy $\pi^i$ that maximizes the expected cumulative discounted reward:

$$J^i(\pi^i) = \mathbb{E}\left[\sum_{t=0}^{\infty} \gamma^t r_t \middle| \pi^i, \pi^{-i}\right], \qquad (1)$$

where $\pi^{-i}$ denotes the policies of all agents other than agent $i$, and the expectation is over the trajectories induced by the policies and the environment dynamics.

In our setting, although the reward function $R$ is known (denoted as original reward function $R_{\text{ori}}$), it is challenging for agents to learn optimal policies due to its sparsity or complexity. This difficulty can lead to slow convergence or suboptimal performance for traditional reinforcement learning algorithms.

## 4. Method

To address the challenges posed by sparse or complex reward functions in multi-agent environments, we introduce the **Multi-phase Human Feedback Markov Game (MHF-MG)** as a tuple, $\langle \mathcal{N}, \mathcal{S}, \mathcal{A}, P, R, \gamma, \mathcal{U}, \pi^h \rangle$. Compared to a standard Markov Game, the added $\mathcal{U}$ denotes the set of possible human utterances or feedback messages. $\pi^h$ represents the human's policy. In this framework, the agents interact with both the environment and a human over discrete generations indexed by $k = 0, 1, \ldots, K - 1$. At each generation $k$, agents collect experiences by interacting with the environment using their current policies $\pi_k^i$.

Each generation $k$ consists of hundreds of iterations; each iteration comprises tens of episodes, and each episode spans on the order of hundreds of environment time steps $t$, depending on the specific environment. This setup allows agents to gain substantial experience within a generation before receiving human feedback. The human possesses a policy $\pi^h$, which may be sub-optimal but is assumed to be initially superior to the agents' policies. This human policy provides valuable guidance that can accelerate the agents' learning. The human observes the agents' behaviors and

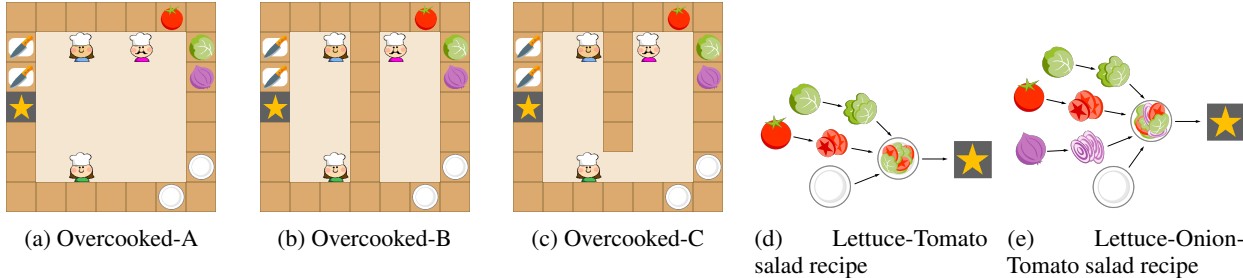

(a) Overcooked-A   (b) Overcooked-B   (c) Overcooked-C   (d) Lettuce-Tomato salad recipe   (e) Lettuce-Onion-Tomato salad recipe

*Figure 2.* The Overcooked Environment. (a)-(c) The three different kitchen layouts with increasing difficulty: (a) Overcooked-A offers ample movable space; (b) Overcooked-B forces agents divided on both sides to cooperate due to the partitioned kitchen; (c) Overcooked-C has less movable space compared to A. (d)-(e) The two salad recipes: In both recipes, the corresponding chopped foods must be combined on a single plate and delivered. To facilitate training, we use macro-actions based on (Xiao et al., 2022), where the agents' actions are simplified. More details refer to Section 5.

generates utterances $u_k \in \mathcal{U}$ at each generation $k$, offering feedback based on the discrepancy between their own policy and the agents' current policies.

We model the human's utterances as a mapping $f$ from the human's policy and the agents' policies to the set of possible utterances:

$$u_k = f\left(\pi^h, \pi_k^1, \pi_k^2, \ldots, \pi_k^N\right), \tag{2}$$

where $f$ captures how the human generates feedback by comparing their policy with those of the agents. The utterances may include specific action recommendations, strategic advice, or corrections aimed at guiding the agents toward better performance. The agents parse the human's utterance $u_k$ to extract actionable information. This process may involve natural language understanding techniques, potentially leveraging large language models (LLMs) to interpret the feedback accurately. Based on the parsed feedback, each agent adjusts its reward function to incorporate the human's guidance. For agent $i$, the updated reward function at generation $k + 1$ becomes:

$$R_{k+1}^i(s, \mathbf{a}) = R(s, \mathbf{a}) + R_{\text{hf}_k}^i(s, \mathbf{a}, u_k), \tag{3}$$

where $R_{\text{hf}_k}^i$ represents the reward adjustment derived from the human's feedback $u_k$ at generation $k$. This adjustment modifies the reward signal to encourage behaviors aligned with the human's guidance, effectively reshaping the agents' learning objectives.

The agents then update their policies for the next generation by optimizing the expected cumulative reward under the new reward function $R^{i,k+1}$. Formally, the policy update for agent $i$ is given by:

$$\pi_{k+1}^i = \arg\max_{\pi^i} \mathbb{E}\left[\sum_{t=0}^{\infty} \gamma^t R_{k+1}^i(s_t, \mathbf{a}_t) \middle| \pi^i, \pi_k^{-i}\right], \tag{4}$$

where $\pi_k^{-i}$ denotes the policies of all other agents at generation $k$, and the expectation is taken over the distribution of

trajectories induced by the policies and the environment dynamics. This iterative process continues across generations, with agents repeatedly interacting with the environment, receiving human feedback, and updating their reward functions and policies accordingly. The inclusion of human feedback helps agents navigate the challenges of sparse or complex reward functions by providing additional signals that highlight desirable behaviors and strategies. In this paper, to minimize human involvement, we limit agent-human interactions to at most five times throughout the entire training process, enabling efficient learning with minimal guidance.

### 4.1. Agent to Human Interaction

In the MHF-MG, agents periodically interact with the human to receive feedback that guides their learning process. This interaction is initiated by the agents, who decide when to seek human input based on specific criteria. The primary mechanism for this interaction is through the generation of rollout trajectories, which approximate the agents' current policy performance.

**Rollouts Generation and Communication.** During each generation $k$, agents interact with the environment using their current policies $\pi_k^i$, collecting substantial experience over multiple iterations. However, due to the complexity or sparsity of the original reward function $R$, agents may still face difficulties in identifying efficient strategies or converging quickly to optimal policies.

To address these challenges, agents periodically seek human feedback based on predefined criteria. Specifically, after every fixed number of training episodes, agents temporarily pause their training and generate evaluation rollouts. These rollouts consist of $X$ independent trajectories collected under the fixed joint policy $\pi^k$, defined formally as:

$$\tau_k^{(x)} = \left\{\left(s_t^{(x)}, \mathbf{a}_t^{(x)}, r_t^{(x)}\right)\right\}_{t=0}^{H-1}, \quad x = 1, \ldots, X, \tag{5}$$

where $H$ is the time horizon, $s_t^{(x)}$ denotes the state at time $t$, $\mathbf{a}_t^{(x)}$ is the joint action chosen by all agents in trajectory $x$, and $r_t^{(x)}$ is the resulting environmental reward. These rollouts allow human observers to assess current agent behaviors and provide structured, actionable feedback for subsequent policy improvements.

**Approximation of Policy Performance via Rollouts.** Consider a stochastic game in which all agents follow stationary joint policies. At generation $k$, the joint policy $\pi^k$ combined with the environment dynamics induces a distribution over state-action-reward trajectories. Our goal is to formally justify that empirical estimates obtained from multiple collected rollouts reliably approximate the true performance $J(\pi^k)$. Given the empirical trajectories $\tau_k^{(m)}$ defined previously, we define each rollout's discounted return as:

$$G_H^{(x)} = (1 - \gamma) \sum_{t=0}^{H-1} \gamma^t r_t^{(x)}, \quad x = 1, \ldots, X, \quad (6)$$

and consider the empirical performance estimator:

$$\hat{J}_{X,H}(\pi^k) = \frac{1}{X} \sum_{x=1}^{X} G_H^{(x)}. \quad (7)$$

Under the Strong Law of Large Numbers (Kolmogorov, 1933; Durrett, 2010), as $M \to \infty$, this estimator converges almost surely to the finite-horizon expected discounted return:

$$\hat{J}_{X,H}(\pi^k) \xrightarrow[X \to \infty]{a.s.} J_H(\pi^k),$$
$$\text{where} \quad J_H(\pi^k) = (1 - \gamma) \sum_{t=0}^{H-1} \gamma^t \, \mathbb{E}[r_t]. \quad (8)$$

Furthermore, when horizon length $H$ grows towards infinity, the finite-horizon discounted return $J_H(\pi^k)$ converges to the true discounted return $J(\pi^k)$. Thus, we have:

**Proposition 4.1** (Performance Estimation). *Under Assumption A.1 and Assumption A.4, assuming bounded rewards and given $X$ independent rollouts of length $H$ collected under policy $\pi^k$, the empirical estimator converges almost surely to the true discounted expected return:*

$$\hat{J}_{X,H}(\pi^k) \xrightarrow[\substack{X \to \infty \\ H \to \infty}]{} J(\pi^k) \quad (a.s.) \quad (9)$$

Leveraging Proposition 4.1, the multiple rollout trajectories serve as reliable empirical estimates of the policy performance $J(\pi^k)$. Human evaluators use these rollouts to effectively assess agent behavior and provide targeted feedback $u_k = f(\tau_k; \pi^h)$. This assumption that empirical rollout data accurately reflects policy-induced distributions is analogous to common practices in offline reinforcement learning and policy evaluation contexts (Levine et al., 2020; Kumar et al., 2019).

## 4.2. Human to Agents

**Feedback Parsing.** Our method employs a Large Language Model (LLM), denoted as $\mathcal{M}$, to parse the human feedback $u_k$ received at generation $k$ and assign it either to specific agents or to all agents collectively. This parsing process is mathematically represented as $u_k^i, u_k^{\text{all}} = \mathcal{M}(u_k, N)$, where $N$ is the number of agents, $u_k^i$ is the feedback assigned to agent $i$, and $u_k^{\text{all}}$ is the feedback applicable to all agents. This approach ensures that each agent receives relevant instructions or corrections based on the human input. The detailed prompts used for guiding the LLM in this parsing process are provided in the Appendix D.4.

**Generating New Reward Function** The new reward function $R_k^i$ for the current generation involves selecting and parameterizing predefined function templates based on human feedback. For each agent $i$, the new reward function for the current generation $k$ is generated as follows:

$$R_k^i(s, a) = \mathcal{M}(F, u_k^i, u_k^{\text{all}}, \boldsymbol{e}), \quad (10)$$

where $F$ is a set of predefined function templates, $u_k^i$ is the parsed feedback for agent $i$ at generation $k$, and $\boldsymbol{e}$ are the entities based on the environment states.

**Predefined Function Templates.** In our framework, we define a set of predefined reward function templates $F$ that can be parameterized based on human feedback and the specific entities within the environment, as shown in Figure 1. These templates enable the system to systematically generate reward functions aligned with human intentions, facilitating efficient policy updates in response to feedback. The templates capture common interaction patterns such as distance-based rewards that encourage agents to minimize their distance to target entities, action-based rewards that incentivize specific actions, and status-based rewards that reward agents for achieving certain states of the environment. For instance, given the human feedback "I think the red chef needs to be responsible for getting the onion" and the LLM will select the distance-based reward template and parameterize it as:

$$R_k^i(s, a) = -\|s[\text{Agent1.pos}] - s[\text{Onion.pos}]\|_2. \quad (11)$$

Here, the $s[\text{Agent1.pos}]$ and $s[\text{Onion.pos}]$ are the relevant entities of the observation vector, which encourages Agent 1 (The red chef in the rollout video) to minimize its distance to the onion, thus aligning its behavior with the desired objective. This structured approach allows agents to interpret and act upon multi-fidelity human feedback effectively. Detailed formulations of these reward templates and additional examples are provided in the Appendix D.2.

**Reward Function for the next generation.** At the end of the processing of the human feedback in generation $k$, we conclude the final reward function for each agent to a

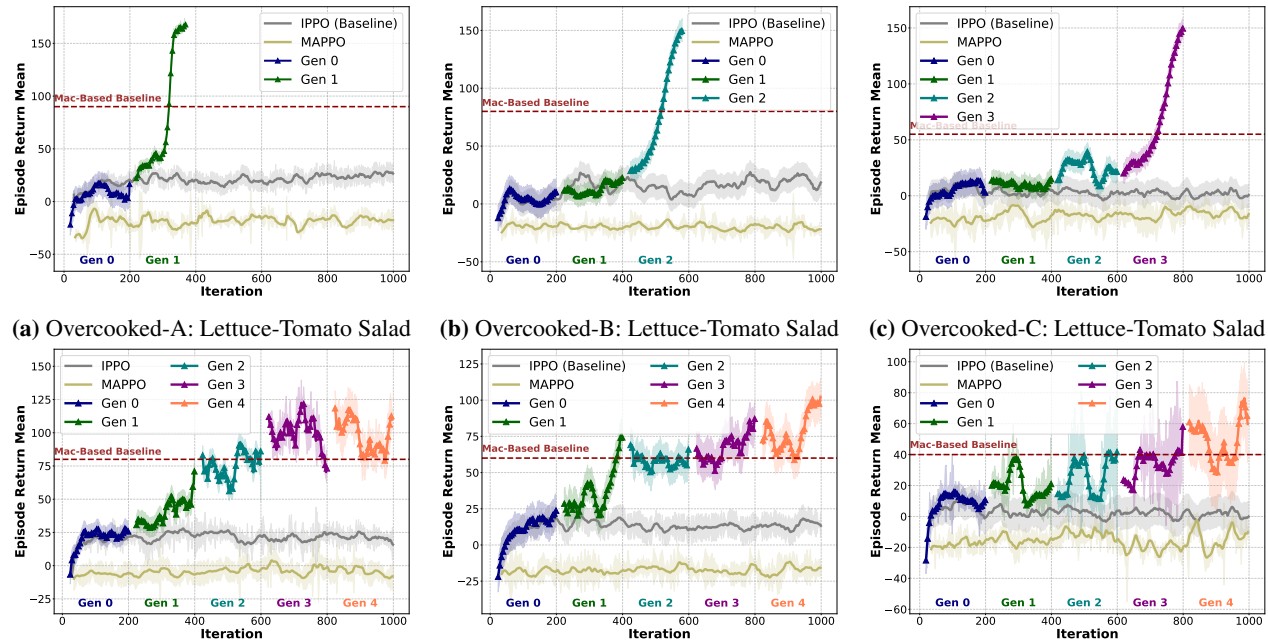

**(a)** Overcooked-A: Lettuce-Tomato Salad  **(b)** Overcooked-B: Lettuce-Tomato Salad  **(c)** Overcooked-C: Lettuce-Tomato Salad

**(d)** Overcooked-A : Lettuce-Onion-Tomato  **(e)** Overcooked-B : Lettuce-Onion-Tomato  **(f)** Overcooked-C : Lettuce-Onion-Tomato

*Figure 3.* Performance comparison of M³HF against baseline methods across different Overcooked environments and recipes. The plots show the mean episode return over 1000 training iterations (approximately 25k episodes) for (a-c) Lettuce-Tomato salad recipe and (d-f) Lettuce-Onion-Tomato salad recipe in Overcooked layouts A, B, and C, respectively. M³HF consistently outperforms the baseline methods (Mac-based Baseline, IPPO, MAPPO) across all scenarios, with performance improvements becoming more pronounced in more complex environments and recipes. Vertical lines indicate the start of each generation where human feedback is incorporated. All experiments are run with three random seeds, and the shaded areas represent the standard deviation.

weighted combination of the base reward and the consistency weight:

$$\hat{R}^i_{k+1}(s,a) = \sum_j w_{i,j} \cdot R^i_j(s,a) \,, \forall R^i_j \in P_i \quad (12)$$

Here, $\hat{R}^i_{k+1}(s,a)$ denotes the final reward function for agent $i$ after processing the $k$-th generation of human feedback, and it will be used for the next generation $k+1$ for the policy training and await the subsequent rollout generation and human interaction, as outlined in Algorithm 1.

One main challenge is how to set the weights $w_{i,j}$ which can effectively balance different reward components and adapt to changing human feedback. To address this challenge, we employ weight decay and performance-based adjustment to optimize the weights $w_{i,j}$.

### 4.3. Weight Decay and Performance-based Adjustment

The straightforward way to adjust weights is based on a simple weight decay mechanism and performance feedback. When generating a new reward function, we add it to the pool $P_i$, then set an initial weight, $w_{i,m} = \frac{1}{|P_i|+1}$. Then, we apply a decay to existing weights of the former reward

functions:

$$w_{i,m} = w_{i,m} \cdot \alpha^{M-m}, \forall m \in 1, \dots, M-1, \quad (13)$$

where $\alpha \in (0,1)$ is a constant decay factor. We then normalize all weights by using

$$w_{i,m} = \frac{w_{i,m}}{\sum w_{i,m}}, \forall m \in 1, \dots, M. \quad (14)$$

Additionally, we introduce a performance-based adjustment rule that compares the agent's performance under the original reward function $R^i_{\text{ori}}$ across consecutive generations. We calculate $r_{\text{ori}}{}^i_{k+1} - r_{\text{ori}}{}^i_k$, where $r_{\text{ori}}{}^i_{k+1}$ is the performance of the policy trained using the new reward function $\hat{R}^i_{k+1}(s,a)$ (after processing human feedback in generation $k$) when evaluated on $R^i_{\text{ori}}$, and $r_{\text{ori}}{}^i_k$ is the performance of the policy trained using the previous reward function $R^i_k(s,a)$ (before processing human feedback in generation $k$) when evaluated on $R^i_{\text{ori}}$. If this difference is positive, it indicates that the new reward function leads to improved performance on the original task. Otherwise, it suggests that the new reward function may be detrimental to the agent's performance on the original task. We then adjust the weight of the newest

reward function component $w_{i,m}$ as follows:

$$w_{i,m} = \begin{cases} w_{i,m} + \beta, & \text{if } r_{\text{ori}}{}_{k+1}^i - r_{\text{ori}}{}_k^i > 0, \\ \max(0, w_{i,m} - \beta), & \text{otherwise,} \end{cases}$$
(15)

where $\beta$ is a small adjustment factor. This approach allows for the dynamic adjustment of the reward function pool, emphasizing recent human feedback while maintaining a diverse set of reward components and adapting to performance changes.

### 4.4. Analysis of the Low-Quality Feedback

We now examine the robustness of our proposed M³HF framework under the scenario where human feedback of mixed quality—including noisy, irrelevant, or erroneous instructions—is provided to the agents. Such situations naturally arise due to confusion, limited human domain expertise, misinterpretation of agent behaviors, or misunderstanding of the task objectives. Under these conditions, it is crucial for a robust algorithm to clearly mitigate the negative impact from low-quality feedback signals, while consistently leveraging high-quality feedback to enhance learning. Formally, at each generation $k$, we integrate rewards by forming a weighted combination of the original task-based reward and the accumulated human-derived feedback rewards:

$$\widehat{R}_k = \sum_{m=0}^{k} w_m^k R_m,$$
(16)

where we define $R_0 = R_{ori}$ as the original reward function and $R_{m>0} = R_{human}$ as human-generated feedback rewards. Correspondingly, the weights $w_m^k$ are dynamically adjusted as described in Equation 15, explicitly accounting for observable agent performance improvements or degradations.

Due to the design of our weighting mechanism, negative or unhelpful feedback rapidly loses influence, as their corresponding weights decrease after any observed dip or stagnation of performance improvements. Conversely, helpful feedback continuously guides performance upwards, maintaining substantial influence through increased weighting in the combined reward function. We formalize this intuition into the following robustness result:

**Proposition 4.2** (Robustness to Low-Quality Human Feedback). *Under Assumption A.3 (Performance Estimation Accuracy) and Assumption A.2 (Learning Algorithm Convergence), for any given integer $K \geq 1$ and any arbitrary sequence of human feedback rewards $(R_k)_{k=1,2,\ldots,K}$, the performance improvement satisfies:*

$$J_{\text{ori}}(\pi_K) - J_{\text{ori}}(\pi_0) \geq \sum_{j=1}^{n(K)} \Delta r_{i_j} - (\delta - \epsilon),$$
(17)

*where $\delta$ is a bounded, positive constant independent of $K$, and each $i_j$ denotes a generation index at which the received human feedback reward is beneficial (i.e., yields a positive increment $\Delta r_{i_j} > 0$).*

Intuitively, Proposition 4.2 formalizes the notion that our algorithm accumulates the positive contributions of helpful feedback over multiple rounds of interaction. In contrast, its performance can suffer at most a single bounded degradation, reflecting the limited and transient influence of the most recent faulty feedback. Importantly, this robustness property ensures stable long-term learning dynamics even when human inputs are imperfect or mixed quality. We provide a detailed proof of this proposition in Appendix B.

## 5. Experiment

In our experiment, we aim to address three key questions: **Q1.** What is the overall performance of M³HF compared to current state-of-the-art methods? **Q2.** To what extent does multi-quality human feedback impact the performance of M³HF within the same environment? **Q3.** Can Vision-Language Models (VLMs) serve as a scalable and effective alternative to human feedback in M³HF? In all experiments involving language-driven feedback parsing, we use the LLM gpt-4o-2024-11-20 (OpenAI, 2024).

**Environment - Macro-Action-Based Overcooked**, as shown in Figure 2. In our experiments, we utilize a challenging multi-agent environment based on the Overcooked game (Wu et al., 2021; Xiao et al., 2022), where three agents must learn to cooperatively prepare a correct salad and deliver it to a designated delivery cell. We followed the work of Xiao et al. (2022), where agents operate using macro-actions derived from primitive actions. These macro-actions facilitate effective navigation and interaction within the environment but also introduce complexities in learning optimal policies due to the increased action space. Each agent observes only the *positions* and *statuses* of entities within a $5 \times 5$ square centered on itself, introducing partial observability and heightening the coordination challenge. The agents will only receive a significant reward for delivering the correct salad (+200) and punishment if they deliver the wrong salad or food (-50). During training, each generation consists of 200 iterations; each iteration runs 25 episodes of up to 200 timesteps. For more details about the environment setting, please refer to the Appendix C.

**Baselines.** We evaluate against three strong multi-agent reinforcement learning approaches: The **MAPPO** (Yu et al., 2022), **IPPO** (De Witt et al., 2020), and a **Macro-Action-Based Baseline** from Xiao et al. (2022). Our own framework adopts IPPO as the backbone algorithm, while the macro-action baseline is the average performance of the two best-performing methods in Xiao et al. (2022), namely Mac-

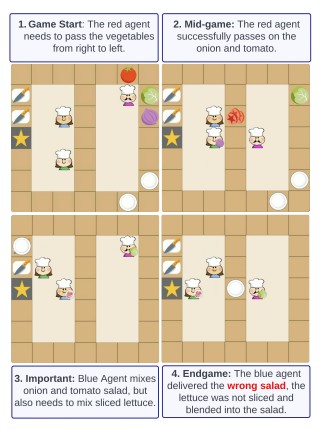
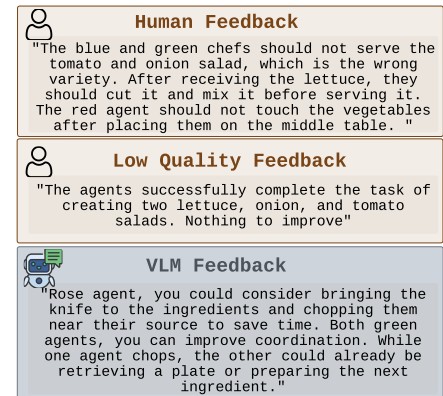
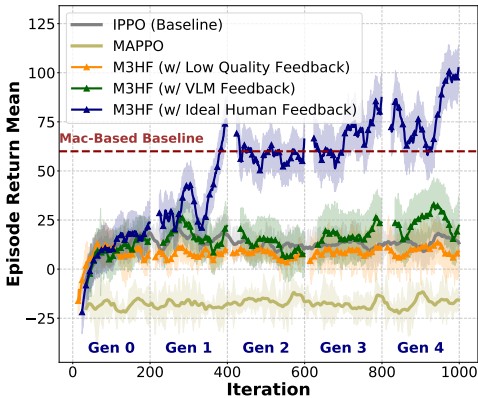

**(a)** Example rollout in Generation 3    **(b)** Feedback example from different source    **(c)** Overcooked-B : Lettuce-Onion-Tomato Salad

*Figure 4.* Impact of Mixed-Quality Feedback on Agent Performance. (a) An example is the rollout in Generation 3, where agents exhibit suboptimal behavior due to poor coordination and inefficient task execution. (b) Low-quality feedback provided to the agents, inaccurately stating that they successfully completed the task and offering no constructive guidance for improvement. (c) Performance comparison on Overcooked-B with the Lettuce-Onion-Tomato salad recipe under mixed quality feedback conditions.

IAICC and Mac-CAC, over 25,000 training episodes. Further details on these baselines can be found in Appendix D.

**Experiment Results for Question 1: Overall Performance of M³HF** Figure 3 demonstrates the superior performance of M³HF compared to SOTA baselines across various Overcooked environments and recipes. Our method consistently outperforms Mac-based Baseline, IPPO, and MAPPO in all scenarios, maintaining a substantial performance advantage across different levels of task complexity. The method exhibits accelerated learning, particularly in early training stages, and achieves higher asymptotic performance levels. Notably, in the simpler recipe setting, Figure 3a, 3b and 3c, M³HF converges to the optimal performance less than five rounds of interaction, showcasing the method's exceptional efficiency in more straightforward settings. The method's robustness is evident as we move to more complex environments. In the challenging Layout C (Figure 3c and 3f), M³HF maintains its effective performance advantage, particularly outperforming its backbone algorithm IPPO. This consistent superiority across varying complexity levels underscores M³HF's effectiveness and adaptability in diverse multi-agent scenarios.

In addition, it is important to note that the baseline MAPPO employs a shared policy among agents alongside a centralized value function, while IPPO utilizes independent policies. We have observed that IPPO often achieves better results in highly coordination-intensive scenarios such as Overcooked, potentially due to reduced interference among agents during policy training. Prior studies (De Witt et al., 2020; Yu et al., 2022) similarly report that IPPO can match or outperform MAPPO even without centralized critics.

**Experiment Results for Question 2: Impact of Mixed-Quality Human Feedback** We evaluated our method when facing low-quality feedback by simulating such feedback at each generation. For example, in the rollout shown in Figure 4a, agents exhibited suboptimal behavior due to poor coordination. Despite this, the low-quality feedback inaccurately stated, "The agents successfully complete the task of creating two lettuce, onion, and tomato salads. Nothing to improve," as depicted in Figure 4b. When training with this irrelevant or erroneous feedback, the agents' performance, illustrated in Figure 4c, remained only slightly below that of the baseline IPPO algorithm and did not degrade significantly. This outcome supports Proposition 4.2, demonstrating that M³HF effectively mitigates the impact of unhelpful feedback through its weight adjustment mechanisms. Even with mixed-quality human input, the framework maintains performance close to the backbone algorithm, showcasing its resilience to low-quality guidance.

**Experiment Results for Question 3: VLM-based Feedback Generation** We explore the potential of VLMs as an alternative to human feedback. The VLM is given the same video rollouts that humans would observe, sampled at a rate of 1 frame per second. Using all sampled frames and a prompt asking for feedback (detailed in Appendix D.4), the VLM generates feedback to the training agents. In our implementation, we leverage `Gemini-1.5-Pro-002` (Reid et al., 2024), which is chosen for its multimodal understanding capability across a long context. We showcase an example of VLM feedback in Figure 4b alongside the human feedback. Here, the feedback provided by the VLM resembles human-like style but lacks specificity on critical issues, which, in this case, are "wrong variety", "cut it and

*Table 1.* Ablation results comparing feedback parsing and weight adjustments. Overcooked-B: Lettuce-Onion- Tomato Salad scenario.

| Method | Average Return (Mean ± Std) |
| --- | --- |
| Raw Feedback (w/o parsing) | 45.3 ± 5.2 |
| LLM Parsing Only (no weight adj.) | 68.7 ± 4.1 |
| Full M$^3$HF (LLM parsing + weight adj.) | **102.7 ± 10.8** |

*Table 2.* Performance comparison between single-phase and multi-phase feedback methods. Overcooked-B: Lettuce-Onion- Tomato Salad scenario.

| Method | Average Return (Mean ± Std) |
| --- | --- |
| Single-phase feedback (initial only) | 43.1 ± 10.3 |
| Multi-phase feedback (M$^3$HF) | **102.7 ± 10.8** |

mix it before serving it". Instead it offers vague suggestions like "improve coordination", which is hard to translate into reward design. This limitation is indicative of the VLM's current inability to perform complex reasoning across images. As a result, when plugged into our M$^3$HF framework, the VLM feedback method does not yield much benefit, as shown in Figure 4c. Nonetheless, we expect improved performance with future advancements in VLM.

**Ablation 1: Impact of Feedback Parsing and Weight Adjustment.** We compare our complete proposed feedback integration pipeline ("Full M$^3$HF") against two variants: *Raw Feedback*, which directly translates human instructions into rewards without parsing or structured adjustment, and *LLM Parsing Only*, which parses human feedback into structured reward functions without performance-based weight adjustment. The results are summarized in Table 1. The results indicate that structured parsing of feedback impacts considerably performance, increasing average returns significantly compared to raw human feedback. Furthermore, adding weight adjustment mechanisms based on agent performance improvements further amplifies policy learning efficiency, underscoring the importance of combining structured parsing and dynamic reward weighting.

**Ablation 2: Single-phase vs. Multi-phase Feedback.** We compared our proposed multi-phase feedback collection methodology against a single-phase scenario (feedback provided only once at the start of training). Results are summarized in Table 2. Table 2 demonstrates superior performance for our multi-phase framework over single-phase feedback, highlighting the effectiveness of iterative, incremental feedback in enabling agents' policy refinement across multiple training stages.

**Ablation 3: Comparison to Intrinsic Reward Methods.** We further evaluated the effectiveness of M$^3$HF relative to intrinsic reward-based methods, specifically comparing against the IRAT method (Wang et al., 2022). As Overcooked lacks built-in intrinsic rewards, we manually constructed three variants of intrinsic reward functions (denoted

*Table 3.* Comparison of intrinsic reward-based methods with our M$^3$HF on Overcooked-B: Lettuce-Onion- Tomato Salad task. Evaluation at training iterations 400, 600, 800, and 1000 (corresponding to Generations 1–4 in M3HF). Mean and standard deviation reported over three seeds.

| Algorithm | Iter. 400 | Iter. 600 | Iter. 800 | Iter. 1000 |
| --- | --- | --- | --- | --- |
| IPPO (base) | 19.2 ± 4.5 | 23.1 ± 2.7 | 23.2 ± 3.3 | 27.4 ± 4.9 |
| IRAT-rw_1 | 68.9 ± 10.1 | 52.5 ± 11.3 | 78.2 ± 14.5 | 94.9 ± 10.7 |
| IRAT-rw_2 | 1.1 ± 2.1 | 9.3 ± 11.4 | 16.0 ± 8.1 | 34.5 ± 14.0 |
| IRAT-rw_3 | 10.8 ± 9.1 | 17.3 ± 10.6 | 21.3 ± 8.7 | 33.8 ± 9.9 |
| M$^3$HF (Ours) | **164.8 ± 1.2** | — | — | — |

as rw_1, rw_2, rw_3) reflecting progressively stronger coordination requirements:

- rw_1: Rewards an agent whenever it picks up or chops any ingredient.

- rw_2: Rewards the agent upon successfully reaching the knife after picking up an ingredient.

- rw_3: Rewards the agent only upon successfully chopping an ingredient after reaching the knife, approximating an optimal coordination strategy.

The results summarized in Table 3 show that while intrinsic reward methods (IRAT variants) typically enhance performance over vanilla IPPO, they still remain significantly inferior to M$^3$HF, especially in the early stages of training. This is largely due to IRAT's reliance on predefined reward structures generated prior to observing actual policy behaviors, resulting in suboptimal coordination patterns (e.g., multiple agents crowding the same object). In contrast, M$^3$HF utilizes human feedback derived from observed rollouts, precisely identifying and targeting coordination failures, enabling agents to quickly improve policy behaviors and achieve superior cooperative results.

## 6. Conclusion

In this paper, we introduced M$^3$HF, a novel framework for MARL that incorporates multi-phase human feedback of mixed quality to address the challenges of sparse or complex reward signals. By extending the Markov Game to include human input and leveraging LLMs to parse and integrate human feedback, our approach enables agents to learn more effectively. Empirically, M$^3$HF outperforms strong baselines, particularly in scenarios with increasing complexity. Our findings highlight the potential of integrating diverse human insights to enhance multi-agent policy learning in a more accessible way.

## Impact Statement

This research introduces $M^3HF$, a framework that enables multi-agent reinforcement learning systems to learn effectively from human feedback of varying quality. By allowing non-expert humans to provide meaningful feedback to AI systems, $M^3HF$ democratizes the development of multi-agent systems while making them more robust to real-world situations where perfect expert guidance may not be available. This could accelerate the deployment of collaborative AI systems in areas such as healthcare, manufacturing, and emergency response, where multiple agents need to coordinate while incorporating human domain knowledge. While this increased accessibility could lead to broader adoption, we acknowledge the importance of appropriate oversight and encourage future work to explore necessary safeguards for such systems.

## Acknowledgment

This work was supported by the Engineering and Physical Sciences Research Council [grant number EP/Y003187/1, UKRI849]. This work was also supported in part by NSF grant IIS-2046640 (CAREER).

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

## A. Assumptions

We make the following assumptions to facilitate the analysis:

**Assumption A.1** (Bounded Rewards). All reward functions, including the original reward $R_{\text{ori}}^i$, the true human feedback reward $R_{\text{true}k}^i$, and the noisy human feedback reward $R_{\text{hf}k}^i$, are uniformly bounded:

$$|R(s,a)| \leq R_{\max}, \quad \forall s, a. \tag{18}$$

This assumption is standard in reinforcement learning to ensure stability and convergence (Sutton, 2018).

**Assumption A.2** (Learning Algorithm Convergence). Given a fixed reward function, the learning algorithm converges to a policy that is $\epsilon$-optimal with respect to the expected cumulative reward under that reward function:

$$\lim_{t \to \infty} \mathbb{E}[J_R^i(\pi_t^i)] \geq J_i^R(\pi_*^i) - \epsilon, \tag{19}$$

where $J_R^i(\pi)$ is the expected cumulative reward for agent $i$ under policy $\pi$ and reward function $R$, and $\pi_*^i$ is the optimal policy for agent $i$ under $R$.

**Assumption A.3** (Performance Estimation Accuracy). The estimate of the performance difference $\Delta r_k^i = J_{\text{ori}}^i(\pi_{k+1}^i) - J_{\text{ori}}^i(\pi_k^i)$ accurately reflects the true change in expected cumulative reward under the original reward function $R_{\text{ori}}^i$ between generations $k$ and $k+1$.

This assumption relies on having sufficient samples to estimate the performance difference accurately, which can be ensured through appropriate exploration and sample size.

**Assumption A.4** (Independent Roll-outs). The $X$ rollout trajectories $\{\tau_k^{(x)}\}_{x=1}^X$ collected under the fixed joint policy $\pi_k$ are independent and identically distributed (i.i.d.).

## B. Proof of Proposition 4.2

In this section, we provide a complete and structured proof for Proposition 4.2.

Let $i_j$ denote the generation index at which the incorporation of human feedback explicitly yields a positive policy improvement for the $j$-th time, and let $n(K)$ represent the number of times positive improvements (helpful human feedback) occur until timestep $K$. Formally, each $i_j$ satisfies $\Delta r_{i_j} > 0$, and indices between $i_{j-1}$ and $i_j$ denote interactions with low-quality or unhelpful feedback where $\Delta r_k < 0$.

The proposition explicitly states that for any generation $k$ in the interval $i_{j-1} \leq k < i_j$, the performance satisfies:

$$J(\pi_k) - J(\pi_0) = \sum_{l=1}^{j-1} \Delta r_{i_l}, \quad \text{for} \quad k = i_{j-1},$$

and

$$J(\pi_k) - J(\pi_0) \geq \sum_{l=1}^{j-1} \Delta r_{i_l} - \delta, \quad \text{for all } i_{j-1} < k < i_j,$$

where $\delta$ is a small bounded positive constant defined explicitly later.

The proof proceeds in three steps:

**Step 1: Performance at positive improvement milestones** ($k = i_{j-1}$). By definition of $i_j$, we clearly have:

$$J(\pi_{i_{j-1}}) - J(\pi_0) = \sum_{l=1}^{j-1} \left( J(\pi_{i_l}) - J(\pi_{i_{l-1}}) \right) = \sum_{l=1}^{j-1} \Delta r_{i_l}.$$

**Step 2: Performance between positive milestones** ($i_{j-1} < k < i_j$). For indices between $i_{j-1}$ and $i_j$, human feedback results in negative or zero performance improvement ($\Delta r_k \leq 0$). According to our algorithm and Eq. (15), the weight assigned to any unhelpful feedback reward function is adjusted (reduced), and eventually clipped to zero. Specifically, whenever this negative improvement occurs, the adaptive weighting mechanism significantly reduces or eliminates the influence of this poor-quality feedback. Thus, the previously established optimal policy at timestep $i_{j-1}$ remains largely unchanged. The largest possible degradation from the previous high-performance point is therefore bounded by a constant $\delta > 0$, independent of the total interaction length $K$:

$$J(\pi_k) - J(\pi_{i_{j-1}}) \geq -\delta, \quad \forall i_{j-1} < k < i_j.$$

**Step 3: Combining two results.** Combining the two preceding steps directly, we establish the performance lower bound at any time step $k$ within the interval:

$$\begin{aligned} &J(\pi_k) - J(\pi_0) \\ &= (J(\pi_k) - J(\pi_{i_{j-1}})) + (J(\pi_{i_{j-1}}) - J(\pi_0)) \\ &\geq -\delta + \sum_{l=1}^{j-1} \Delta r_{i_l}. \end{aligned}$$

Since this holds for all intervals of indices, applying it specifically to the final interval (after the last helpful feedback index $n(K)$), we obtain the desired inequality:

$$J(\pi_K) - J(\pi_0) \geq \sum_{j=1}^{n(K)} \Delta r_{i_j} - \delta.$$

Thus, Proposition 4.2 follows directly.

**Explicit bound on the constant $\delta$.** We next explicitly quantify the bounded constant $\delta$ appearing due to one step of degraded performance. Consider the following general lemma:

**Lemma B.1.** *Refer to Puterman (1990) and Lemma 6.2 in Bertsekas & Tsitsiklis (1996), Let $r_1, r_2, r_3$ be three bounded reward functions. Define combined rewards:*

$$\begin{cases} R = (1-p)\,r_1 + p\,r_2, \\ R' = (1-p'-q)\,r_1 + p'\,r_2 + q\,r_3. \end{cases}$$

*Let $\pi$ and $\pi'$ be the optimal policies corresponding respectively to $R$ and $R'$. Then the following performance bound holds, for a discount factor $\gamma \in (0,1)$:*

$$V_{r_1}^{\pi} - V_{r_1}^{\pi'} \le \frac{2}{1-\gamma}\,\|\,R - R'\,\|_\infty$$

$$\le \frac{2}{1-\gamma}\big[|p' + q - p|\,\|\,r_1 - r_3\|_\infty + |p - p'|\,\|\,r_2 - r_3\|_\infty\big].$$

We now explicitly apply this lemma in our scenario. Consider the timestep $k$ immediately after the last beneficial feedback index $i_j$, with:

- $r_1 = R_{ori}$, the original environment reward.

- $r_2 =$ weighted combination of previously beneficial feedback.

- $r_3 = R_k$, the current negative feedback reward.

Then, we have:

$$V_{R_{ori}}^{\pi_k} - V_{R_{ori}}^{\pi_{i_j}} \le \frac{2}{1-\gamma}\big[|p' + q - p|\|R_{ori} - R_k\|_\infty + |p - p'|\|r_2 - R_k\|_\infty\big]$$

$$(20)$$

By the current weight updating scheme (Eqs. 15), we explicitly have:

- $q = \frac{1/(2+j)}{1+1/(2+j)} = \frac{1}{3+j}$, and

- $0 \le p - p' \le p - \alpha^j p = p(1 - \alpha^j)$.

Noting that the feedback probability $p$ is bounded (e.g. $0 \le p \le 1$) and that all reward functions satisfy $\|r\|_\infty \le R_{\max}$, we obtain a uniform upper bound for $\delta$. In particular,

$$\delta \le \frac{2}{1-\gamma}\Big[\big|\tfrac{1}{3+j} - (1-\alpha^j)p\big|\,\|R_{ori} - R_k\|_\infty + p\,(1-\alpha^j)\,\|r_2 - R_k\|_\infty\Big].$$

$$(21)$$

Since $p \le 1$ and $\|R_{ori} - R_k\|_\infty, \|r_2 - R_k\|_\infty \le R_{\max}$, the right-hand side is a finite constant independent of the total iteration count $K$. This uniform bound on $\delta$ highlights the robustness of our feedback-integration mechanism even when occasional poor-quality feedback occurs.

This completes the full proof.

## C. Environment Details

In this section, we will introduce the details of the environments we are using. We follow the setting from

**Goal**. Three agents need to learn cooperating with each other to prepare a Tomato-Lettuce-Onion salad and deliver it to the 'star' counter cell as soon as possible. The challenge is that the recipe of making a tomato-lettuce-onion salad is unknown to agents. Agents have to learn the correct procedure in terms of picking up raw vegetables, chopping, and merging in a plate before delivering.

**State Space**. The environment is a 7×7 grid world involving three agents, one tomato, one lettuce, one onion, two plates, two cutting boards and one delivery cell. The global state information consists of the positions of each agent and above items, and the status of each vegetable: chopped, unchopped, or the progress under chopping.

**Primitive-Action Space**. Each agent has five primitive-actions: *up*, *down*, *left*, *right* and *stay*. Agents can move around and achieve picking, placing, chopping and delivering by standing next to the corresponding cell and moving against it (e.g., in Figure 2a, the pink agent can *move right* and then *move up* to pick up the tomato).

**Macro-Action Space**. Here, we first describe the main function of each macro-action and then list the corresponding termination conditions.

- Five one-step macro-actions that are the same as the primitive ones;

- *Chop*, cuts a raw vegetable into pieces (taking three time steps) when the agent stands next to a cutting board and an unchopped vegetable is on the board, otherwise it does nothing; and it terminates when:
    - The vegetable on the cutting board has been chopped into pieces;
    - The agent is not next to a cutting board;
    - There is no unchopped vegetable on the cutting board;
    - The agent holds something in hand.

- *Get-Lettuce*, *Get-Tomato*, and *Get-Onion*, navigate the agent to the latest observed position of the vegetable, and pick the vegetable up if it is there; otherwise, the agent moves to check the initial position of the vegetable. The corresponding termination conditions are listed below:
    - The agent successfully picks up a chopped or un-chopped vegetable;

- The agent observes the target vegetable is held by another agent or itself;
- The agent is holding something else in hand;
- The agent's path to the vegetable is blocked by another agent;
- The agent does not find the vegetable either at the latest observed location or the initial location;
- The agent attempts to enter the same cell with another agent, but has a lower priority than another agent.

- **Get-Plate-1/2**, navigates the agent to the latest observed position of the plate, and picks the vegetable up if it is there; otherwise, the agent moves to check the initial position of the vegetable. The corresponding termination conditions are listed below:

  - The agent successfully picks up a plate;
  - The agent observes the target plate is held by another agent or itself;
  - The agent is holding something else in hand;
  - The agent's path to the plate is blocked by another agent;
  - The agent does not find the plate either at the latest observed location or at the initial location;
  - The agent attempts to enter the same cell with another agent but has a lower priority than another agent.

- **Go-Cut-Board-1/2**, navigates the agent to the corresponding cutting board with the following termination conditions:

  - The agent stops in front of the corresponding cutting board, and places an in-hand item on it if the cutting board is not occupied;
  - If any other agent is using the target cutting board, the agent stops next to the teammate;
  - The agent attempts to enter the same cell with another agent but has a lower priority than another agent.

- **Go-Counter** (only available in Overcook-B, Figure 2 b), navigates the agent to the center cell in the middle of the map when the cell is not occupied, otherwise it moves to an adjacent cell. If the agent is holding an object the object will be placed. If an object is in the cell, the object will be picked up.

- **Deliver**, navigates the agent to the 'star' cell for delivering with several possible termination conditions:

  - The agent places the in-hand item on the cell if it is holding any item;
  - If any other agent is standing in front of the 'star' cell, the agent stops next to the teammate;

- The agent attempts to enter the same cell with another agent, but has a lower priority than another agent.

**Observation Space**: The macro-observation space for each agent is the same as the primitive observation space. Agents are only allowed to observe the *positions* and *status* of the entities within a $5 \times 5$ view centered on the agent. The initial position of all the items are known to agents.

**Dynamics**: The transition in this task is deterministic. If an agent delivers any wrong item, the item will be reset to its initial position. From the low-level perspective, to chop a vegetable into pieces on a cutting board, the agent needs to stand next to the cutting board and executes *left* three times. Only the chopped vegetable can be put on a plate.

**Original Reward Function**: $+10$ for chopping a vegetable, $+200$ terminal reward for delivering a correct salad (like tomato-lettuce-onion or tomato-lettuce salad), $-5$ for delivering any wrong entity, and $-0.1$ for every timestep.

**Episode Termination**: Each episode terminates either when agents successfully deliver a tomato-lettuce-onion salad or reaching the maximal time steps, 200.

## D. Implementation Details

### D.1. Algorithm

In here, we list the complete algorithm, as shown in Algorithm.1.

### D.2. Predefined Reward Function Templates

To effectively incorporate human feedback into the learning process, we define a set of predefined reward function templates $F$ that can be parameterized based on the feedback and entities present in the environment. These templates capture common interaction patterns between agents and their environment, facilitating automatic reward function generation aligned with human intentions.

Firstly, the **distance-based reward** function penalizes the agent proportionally to the Euclidean distance between two entities $e_1$ and $e_2$ within the environment:

$$f_{\text{dist}}(s, a, e_1, e_2) = -\|s[e_1.\text{pos}] - s[e_2.\text{pos}]\|_2, \quad (22)$$

where $s[e_i.\text{pos}]$ denotes the position vector of entity $e_i$ in state $s$, and $\|\cdot\|_2$ represents the Euclidean norm.

Secondly, the **action-based reward** function provides a

**Algorithm 1** M³HF: Multi-agent Reinforcement Learning from Multi-phase Human Feedback of Mixed Quality

---

**Require:** Number of agents $N$, Original Reward Functions $\{R_i^{\text{ori}}\}_{i=1}^N$, Predefined Reward Templates $F$, Environment $E$, Initial Policies $\{\pi^{i,0}\}_{i=1}^N$, Total Generations $K$

**Ensure:** Trained Policies $\{\pi^{i,K}\}_{i=1}^N$

1: Initialize Reward Function Pools $P_i = \{R_i^{\text{ori}}\}$ for each agent $i$

2: **for** generation $k = 0$ to $K - 1$ **do**

3:    **Multi-agent Training Phase**     ▷ **Eq. 4**

4:       **for** each agent $i$ **do**

5:          Train policy $\pi^{i,k}$ using current reward function $\hat{R}_i^k(s, a)$ (Eq. 12)

6:       **end for**

7:    **Rollout Generation**          ▷ **Sec. 4.1**

8:       **if** Periodic evaluation or performance stagnation detected **then**

9:          Generate rollout trajectories $\tau_k = \{(s_t, \mathbf{a}_t, r_t)\}_{t=0}^{H-1}$

10:       **end if**

11:   **Human Feedback Phase**      ▷ **Sec. 4.2**

12:       Human observes $\tau_k$ and provides feedback $u_k$

13:       **Feedback Parsing:**

14:       Use LLM $\mathcal{M}$ to parse $u_k$ and assign feedback to agents:
$u_k^i, u_k^{\text{all}} = \mathcal{M}(u_k, N)$

15:   **Reward Function Update**     ▷ **Sec. 4.3**

16:       **for** each agent $i$ **do**

17:          Generate new reward function from feedback (Eq. 10):
$R_{i,\text{new}} = \mathcal{M}(F, u_k^i, u_k^{\text{all}}, \boldsymbol{e})$

18:          Add $R_{i,\text{new}}$ to reward function pool $P_i$

19:       **end for**

20:       **Weight Update:**

21:    **for** each agent $i$ **do**

22:          Initialize weight for new reward function:
$w_{i,M} = \frac{1}{|P_i|}$

23:          Apply weight decay to existing weights (Eq. 15):
$w_{i,m} = w_{i,m} \cdot \alpha^{M-m}, \forall m \in \{1, \ldots, M-1\}$

24:          Normalize weights:
$w_{i,m} = \frac{w_{i,m}}{\sum_{j=1}^M w_{i,j}}, \forall m \in \{1, \ldots, M\}$

25:          Compute performance difference:
$\Delta r_i = r_{i,k+1}^{\text{ori}} - r_{i,k}^{\text{ori}}$

26:          Adjust weight of newest reward function:
$$w_{i,M} = \begin{cases} w_{i,M} + \beta, & \text{if } \Delta r_i > 0 \\ \max(0, w_{i,M} - \beta), & \text{otherwise} \end{cases}$$

27:          Update final reward function (Eq. 12):
$\hat{R}_i^{k+1}(s, a) = \sum_{m=1}^M w_{i,m} \cdot R_{i,m}(s, a)$

28:       **end for**

29: **end for**

---

reward when the agent performs a specific desired action $a_{\text{desired}}$:

$$f_{\text{action}}(s, a, a_{\text{desired}}) = \mathbb{I}(a = a_{\text{desired}}), \qquad (23)$$

where $a$ is the action taken by the agent, and $\mathbb{I}(\cdot)$ is the indicator function, returning 1 if the condition is true and 0 otherwise.

Thirdly, the **status-based reward** function rewards the agent when an entity $e$ attains a particular desired status $\text{status}_{\text{desired}}$:

$$f_{\text{status}}(s, a, e, \text{status}_{\text{desired}}) = \mathbb{I}(s[e.\text{status}] = \text{status}_{\text{desired}}), \qquad (24)$$

where $s[e.\text{status}]$ represents the current status of entity $e$ in state $s$.

Additionally, we define a **composite reward** function that allows for more nuanced feedback by combining multiple reward components:

$$f_{\text{comp}}(s, a) = \sum_i \lambda_i f_i(s, a), \qquad (25)$$

where $f_i(s, a)$ are individual reward components (e.g., $f_{\text{dist}}, f_{\text{status}}$), and $\lambda_i$ are weighting coefficients that determine the relative importance of each component.

For instance, given the human feedback "Agent 1 needs to get the onion," we might select the distance-based reward template and parameterize it as:

$$R_i(s, a) = -\|s[\text{Agent1.pos}] - s[\text{Onion.pos}]\|_2. \qquad (26)$$

This reward function encourages Agent 1 to minimize its distance to the onion, thus aligning its behavior with the desired objective.

Furthermore, other templates can be incorporated depending on the environmental context and task requirements. For example, a **proximity-based reward** function provides a reward when an agent is within a certain distance $d$ of a target entity:

$$f_{\text{prox}}(s, a, e_1, e_2, d) = \begin{cases} r_{\text{prox}}, & \text{if } \|s[e_1.\text{pos}] - s[e_2.\text{pos}]\|_2 \leq d, \\ 0, & \text{otherwise}, \end{cases} \qquad (27)$$

where $r_{\text{prox}}$ is the reward assigned for being within distance $d$.

A **time-based penalty** can be introduced to encourage efficient task completion:

$$f_{\text{time}}(s, a, t) = -\beta \cdot t, \qquad (28)$$

where $t$ is the current time step, and $\beta$ is a penalty coefficient reflecting the cost of time.

A **success-based reward** provides a reward upon achieving a specific goal condition:

$$f_{\text{success}}(s, a) = \mathbb{I}(\text{goal\_condition\_met}) \cdot r_{\text{success}}, \qquad (29)$$

where $r_{\text{success}}$ is the reward value assigned when the goal condition is met.

An **energy-based penalty** discourages unnecessary expenditure of resources:

$$f_{\text{energy}}(s, a) = -\gamma \cdot \text{energy}(a), \qquad (30)$$

where $\text{energy}(a)$ represents the energy cost associated with action $a$, and $\gamma$ is a scaling factor.

By leveraging these templates, the system can systematically generate reward functions that align with human feedback, enabling agents to adapt their behavior effectively in response to diverse instructions. This approach allows for the incorporation of mixed-quality human feedback into the learning process, enhancing the agents' ability to perform complex tasks in multi-agent environments.

### D.3. Training Details

Our experiments were conducted on a heterogeneous computing cluster running Ubuntu Linux. The hardware configuration included a variety of CPU models, such as Dual Intel Xeon E5-2650, Dual Intel Xeon E5-2680 v2, and Dual Intel Xeon E5-2690 v3. For accelerated computing, we utilized 3 NVIDIA A30 GPUs. The total computational resources comprised 180 CPU cores and 500GB of system memory.

We performed hyperparameter tuning for all baselines to ensure fair comparison. Specifically, for IPPO and MAPPO, we tuned learning rates, batch sizes, and gradient clipping values. For example, we systematically searched over learning rates in {3e-4, 1e-4}, #sgd_iters in {5, 10}, sgd_batch_size in {1024, 5120}, and entropy coefficient in {0.01, 0.05}, ultimately selecting the configurations with the best validation performance. For the macro-action based baseline, we directly adopted the best-performing hyperparameters reported in Mac-based method (Xiao et al., 2022). We will explicitly include these details in our revised manuscript.

*Table 4.* Hyperparameters used in Overcooked-A, B, and C.

| Hyperparameter | M$^3$HF-IPPO | Baseline-IPPO | Baseline-MAPPO |
|---|---|---|---|
| Training Generation | 5 | - | - |
| Training Iterations | 1000 | 1000 | 1000 |
| Training Episodes | 25k | 25k | 25k |
| Learning Rate | 0.0003 | 0.0003 | 0.0003 |
| Training Batch Size | 5120 | 5120 | 5120 |
| Minibatch Size | 1024 | 1024 | 1024 |
| Epochs | 10 | 10 | 10 |
| Discount Factor ($\gamma$) | 0.99 | 0.99 | 0.99 |
| GAE Lambda ($\lambda$) | 0.95 | 0.95 | 0.95 |
| Clip Parameter | 0.2 | 0.2 | 0.2 |
| Value Function Clip Parameter | 10.0 | 10.0 | - |
| Entropy Coefficient | 0.01 | 0.01 | 0.01 |
| KL Coefficient | 0.2 | 0.2 | - |
| Gradient Clipping | 0.5 | 0.5 | 0.5 |

### D.4. Prompts

---

**Prompt 1: FEEDBACK PARSING PROMPT**

Given the following feedback for a multi-agent system in an Overcooked environment, assign the feedback to appropriate agents or to all agents. The system has **{num_agents}** agents.

Feedback: **{Human Feedback}**

The agent_1 is the chef in Green, agent_2 is the chef in Rose, agent_3 is the chef in Blue.

Return your response in the following JSON format:

```
{{
    "agent_0": "feedback for agent 0",
    "agent_1": "feedback for agent 1",
    ...
    "all": "feedback for all agents"
}}
```

Only include keys for agents that receive specific feedback and 'all' if there's general feedback.

---

**Prompt 2: REWARD FUNCTION BUILD PROMPT**

Given the parsed feedback for an agent in an Overcooked environment, select and parameterize a reward function template.

---

The observation space is a 32-length vector as described in the task description.

Parsed Feedback: **{feedback for this agent}**

Observation Space (32-length vector for each agent):
- Tomato: position (2), status (1) (obs[0:2])
- Lettuce: position (2), status (1) (obs[3:5])
- Onion: position (2), status (1) (obs[6:8])
- Plate 1: position (2) (obs[9:10])
- Plate 2: position (2) (obs[11:12])
- Knife 1: position (2) (obs[13:14])
- Knife 2: position (2) (obs[15:16])
- Delivery: position (2) (obs[17:18])
- Agent 1: position (2) (obs[19:20])
- Agent 2: position (2) (obs[21:22])
- Agent 3: position (2) (obs[23:24])
- Order: one-hot encoded (7) (obs[25:32])

Available function templates:
1. Distance-based: -sqrt((agent_x - target_x)**2 + (agent_y - target_y)**2)
2. Action-based: reward for specific actions (e.g., chopping, picking up)
3. State-based: reward for achieving specific states (e.g., holding an item)
4. Time-based: penalty for time taken
5. Combination of the above

Select a template and parameterize it based on the feedback. Return your response as a Python lambda function that takes the observation vector (obs) and action (act) as input.

For example, Distance between agent 1 and tomato :

lambda obs, act: -sqrt((obs[19] - obs[0])**2 + (obs[20] - obs[1])**2) # Distance between agent 1 and tomato

Ensure that your function uses the correct indices from the observation vector as described in the task description.

---

### Prompt 3: VLM FEEDBACK PROMPT

You are an AI assistant helping to manage an Overcooked environment with multiple agents. The task is to prepare and deliver a

{task_name}.
The environment is a 7x7 grid with various objects and {num_agents} agents.

Observation Space (32-length vector for each agent):
- Tomato: position (2), status (1)
- Lettuce: position (2), status (1)
- Onion: position (2), status (1)
- Plate 1: position (2)
- Plate 2: position (2)
- Knife 1: position (2)
- Knife 2: position (2)
- Delivery: position (2)
- Agent 1: position (2)
- Agent 2: position (2)
- Agent 3: position (2)
- Order: one-hot encoded (7)

MA-V1 Actions (index indicates macro action):
0: No operation
1: Move Up
2: Move Right
3: Move Down
4: Move Left
5: Interact (pick up, put down, chop)

You will be provided with a video of the agents' gameplay, which may be lengthy. Your task is to:
1. Identify and summarize the key actions and strategies employed by the agents throughout the gameplay.
2. Provide constructive feedback based on your observations in a single paragraph. Mark this paragraph with [SUGGESTION].

When generating feedback:
- Address specific agents by their color (e.g., Green agent, Rose agent) or position (e.g., agent on the left, agent near the cutting board).
- Focus on aspects of gameplay that could be significantly improved for any or all agents.
- Offer specific, actionable suggestions that can be immediately applied.
- Relate your feedback to the Overcooked environment, tasks, and overall efficiency.
- Prioritize improvements in teamwork, task allocation, or resource management.
- Consider how the suggestions could impact the agents' performance metrics.

Avoid: - Using overly technical jargon or complex explanations.
- Giving vague or general advice not specific to their gameplay.
- Mentioning anything outside the scope of the Overcooked game.
- Using excessive praise or encouragement.

Provide a brief summary of the agents' actions, followed by a single paragraph of feedback marked with [SUGGESTION], addressing the agents directly about their gameplay in the Overcooked environment. Focus on concrete improvements for any or all agents rather than motivational language.

## E. Regarding the Scalability of Our Method

We extended our evaluation to Google Football 5v5 (Kurach et al., 2020; Song et al., 2024), a complex multi-agent benchmark. M3HF continues to outperform standard MARL baselines with the multi-phased human feedback. Full environment details are provided in the Figure 5,6 and their captions.

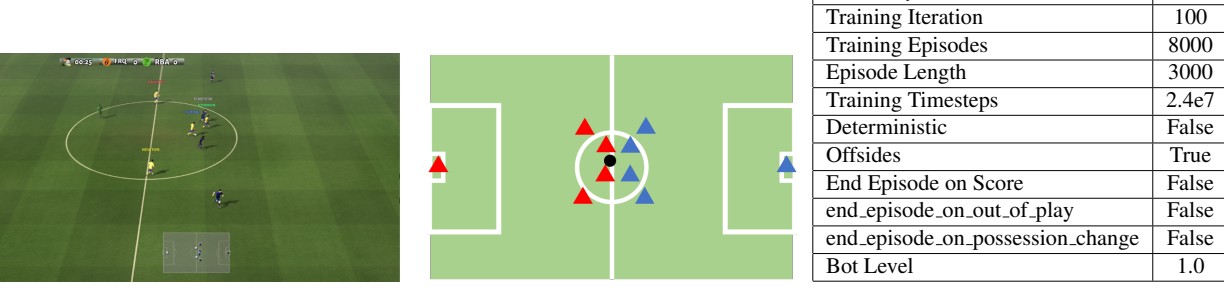

| Total Player Number | 10 |
|---|---|
| Training Iteration | 100 |
| Training Episodes | 8000 |
| Episode Length | 3000 |
| Training Timesteps | 2.4e7 |
| Deterministic | False |
| Offsides | True |
| End Episode on Score | False |
| end_episode_on_out_of_play | False |
| end_episode_on_possession_change | False |
| Bot Level | 1.0 |

**(a)** Rollout Screenshot          **(b)** Start Point          **(c)** 5-vs-5 full-game (5v5) configuration

*Figure 5.* **Google Research Football Environment (5-vs-5 Full Game)**: We evaluate M3HF in the GRF 5-vs-5 HARD Built-in AI scenario, a complex multi-agent benchmark widely used in prior work (Kurach et al., 2020; Song et al., 2024). Each team controls 5 players (10 agents total); **we always control the yellow-shirt (left) team**, which initiates the kickoff. Each episode lasts 3000 steps, with the second half beginning at step 1501. The simulation is accelerated: **1 in-game minute equals 3 real-world seconds**, so a full 90-minute match takes 4.5 minutes. For human feedback collection, we typically present the first attacking phase (60s rollout), corresponding to **20 in-game minutes**. **(a)** shows a rollout snapshot during the opening phase. **(b)** depicts the initial player formation around the center circle. **(c)** presents the full environment configuration. Observations follow a dictionary structure with keys `"obs"` (state features) and `"controlled_player_index"` (agent IDs: 0–4 for yellow team, 5–9 for blue). The action space is `[Discrete(19)]`, where each agent selects one of 19 discrete actions (e.g., pass, sprint, tackle) represented as a one-hot 19D vector.

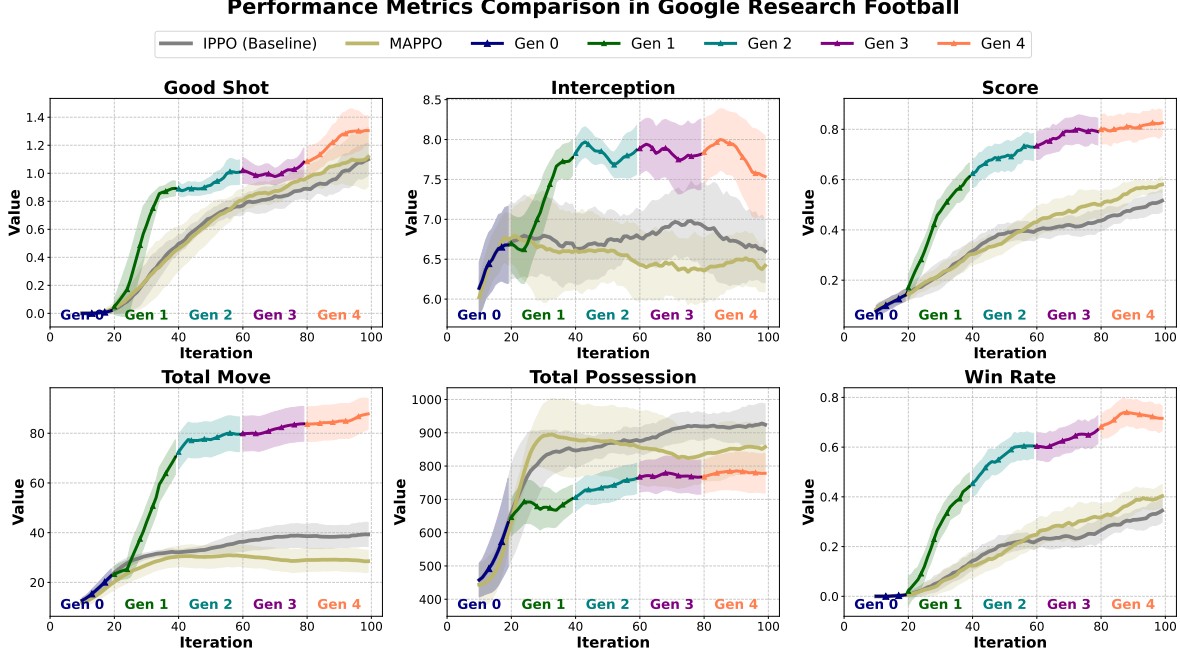

*Figure 6.* **Performance Comparison of M$^3$HF vs. Baselines in GRF 5-vs-5 Full Game.** We evaluate M$^3$HF in the GRF 5v5-HARD setting, a challenging multi-agent benchmark; **Baselines** include IPPO, MAPPO, and macro-action-based MACCS; **Metrics** include Win Rate, Goal Difference, Total Move (indicating spatial coordination), Good Shot (quality shot attempts), and Interception (defensive awareness); **M³HF outperforms all baselines** in most metrics, particularly in Win Rate and coordination-related metrics like Total Move and Good Shot; this is due to multi-phase human feedback that encourages off-ball movement, proactive attacking, and cooperative behavior (e.g., "agents should run into space"); these behaviors result in better scoring opportunities and team coordination; **Conclusion:** M³HF scales effectively to GRF and enables efficient, high-quality policy learning with minimal human input.

