# OpenReview forum: "M³HF: Multi-agent Reinforcement Learning from Multi-phase Human Feedback of Mixed Quality"
_ICML.cc/2025/Conference — ICML 2025 poster_

### Official Review · Reviewer_sgwG · 2025-03-08

**Overall Recommendation:** 4

**Summary:**

In this work, the authors introduce a technique for training agents in MARL settings using human feedback as a substitute for hand-designed reward functions. Specifically, their training pipeline involves (1) collecting human language feedback about a rollout video after a period of training, (2) using an LLM to design a reward function based on the feedback and reward templates, and (3) adjusting weights of reward functions to improve training. They validate their training on Overcooked and find significant improvements in final performance relative to existing MARL baselines.

### Update After Rebuttal

The rebuttal clarified my original confusion between MAPPO and IPPO, and I am satisfied with the additional clarifications regarding theoretical assumptions and proofs.

**Claims And Evidence:**

Claims are generally supported, though I listed specific issues with some claims in later sections of the review.

**Essential References Not Discussed:**

The related works section is comprehensive.

**Experimental Designs Or Analyses:**

- The experimental results show that MAPPO consistently underperforms IPPO, which makes me question the hyperparameter tuning and network architectures used to generate the results. The only difference between MAPPO and IPPO (according to the original MAPPO paper) is the fact that MAPPO has centralized value function inputs, which was found to be broadly helpful. The only other possible difference is that parameter-sharing is used for one implementation but not for the other. I was not able to find details regarding the differences in MAPPO or IPPO in the main text or appendix, so this point needs to be clarified.
- In general, there are other baselines that are simply not presented that attempt to solve the same problem of sparse extrinsic rewards (i.e. the techniques listed in the MARL section of the Related Work). Without comparing to existing baselines, it is unclear how helpful human feedback is relative to intrinsic rewards.

**Methods And Evaluation Criteria:**

Yes, the proposed method and benchmark makes sense for this problem.

**Other Comments Or Suggestions:**

- Multiple typos of word “function” in Figure 1
- In Figure 2’s caption it seems like the descriptions of B and C are swapped?
- The empirical occupancy measure in eq 8 is undiscounted, which means it does not approximate the discounted occupancy measure in eq 6.
- “formal” should be “former” in line 306
- I don’t think “the” should be in the section title of 4.4

**Other Strengths And Weaknesses:**

- The paper reads very well and the diagrams are very clear. Overall, I understand the intuition behind the techniques and why it helps improve performance.

- I disagree with the claim that “designing appropriate reward functions” is a fundamental and significant challenge in MARL (beyond the challenge in single-agent RL). In particular, although naive self-play often converges to suboptimal equilibria, this is often due to the dynamics of learning compatible conventions, not fundamental flaws in the reward function.
    - One possible framing of this work is that human feedback helps align AI behavior to human conventions (though this may come at the cost of stronger self-play performance)
- Despite the framing of the paper as a technique to help multi-agent RL, the technique has little to do with the multi-agent setting other than the fact that Overcooked was chosen as the setting of interest. A difficult single-agent setting could’ve been sufficient for demonstrating the pipeline.
    - Again, framing this as human-AI alignment and conducting human-AI user studies would’ve resolved this concern.
- The “reward templates” require significant domain knowledge and seem to limit the applicability of the technique to new domains.

**Questions For Authors:**

- “Long-term Exploration Failure” for rollout generation is a bit unclear. It seems like the experiments presented only ask humans for feedback every 200 iterations, so how does the long-term exploration failure fit into your final method?

**Relation To Broader Scientific Literature:**

The key contribution of the work is bringing human feedback (though a code-gen LLM) within a MARL training loop as a form of reward shaping. This method appears to be novel to me.

**Theoretical Claims:**

- Proposition 4.1 is valid but irrelevant to the game studied in this setting since the Markov Game is not ergodic. Specifically, we know Overcooked is not ergodic since the “timestep” feature changes at each step (and periodic episode resets violate aperiodicity).
    - I generally felt that the section on “Approximation of Policy Performance via Rollouts” in Section 4.1 was unnecessary and unclear. In particular, it seems like the empirical distribution of states+actions in any single rollout trajectory won’t approximate the “true distribution” but this is easily solved by just having multiple rollouts.
- Proposition 4.2 is based on the assumption that the noise is zero-mean for all states and actions, but this contradicts the premise that this noise is based on “misunderstanding, lack of expertise, or other factors” since the reward function generated by the LLM is deterministic. However, I think the proof still works if we drop the noise term and just state the “true” human feedback reward function is faulty.

---

> ### Author Rebuttal · Authors · 2025-04-01
>
> ## Reply to Reviewer sgwG
> We thank the reviewer for recognizing our pipeline as a "novel" contribution and for the positive comments on clarity and presentation. We address your concerns below.
>
> ---
> ### 1. Regarding the Framing of our work
> Thanks for your thoughtful comments on our paper's scope and framing. We agree that human feedback plays a critical role in improving human-AI alignment. In our revision, we will clarify our motivations and explain how our work contributes to addressing key challenges for human-AI alignment.
>
> ---
> ### 2. Regarding Theoretical Assumptions
> > Proposition 4.1’s relevance:
>
> You are correct that Overcooked is not ergodic due to episode resets and the timestep feature. We included Proposition 4.1 primarily for theoretical completeness, highlighting that rollouts can approximate policy performance under idealized conditions. We will clearly state this limitation and clarify the scope of this proposition in the revision.
> > Proposition 4.2’s noise assumption:
>
> Indeed, as you correctly observe, the reward function generated by the LLM is deterministic, making the original assumption of zero-mean random noise less reflective of our practical setup. Your suggestion—dropping the noise term and explicitly treating the human-generated reward function as potentially faulty—is more consistent with our actual implementation. This perspective strengthens our conclusions, as our adaptive weighting approach does not fundamentally rely on randomness and remains effective even in the presence of deterministic errors. We will explicitly clarify this point in the revision.
>
> ---
> ### 3. Baselines and Intrinsic Reward Comparisons
> >MAPPO vs. IPPO performance:
>
> In our experiments, MAPPO employs a shared policy among agents with a centralized value function, while IPPO uses independent policies. We observed that IPPO often performs better in coordination-intensive scenarios like Overcooked, potentially due to reduced interference during training. Prior works [1,2] similarly report IPPO matching or outperforming MAPPO, even without centralized critics.  We will clarify this in the revision.
>
> > Comparison to intrinsic reward methods:
>
> Our method is orthogonal to intrinsic reward approaches, as M3HF assigns human feedback to each agent’s reward function, effectively serving as a form of tailored intrinsic reward. We also agree that it is valuable to compare intrinsic reward baselines directly; thus, we choose IRAT [3], an intrinsic reward method that combines manually defined reward functions with the original task objective. Since Overcooked lacks pre-built intrinsic rewards, we implement three manually constructed reward variants (rw_1–3) based on increasing levels of agent coordination:
> * rw_1: rewards any pickup or chop of ingredients
> * rw_2: rewards reaching the knife after picking up ingredients
> * rw_3: rewards chopping after reaching the knife and picking up.
> | Iters. |400|600|800|1000|
> |-|-|-|-|-|
> | IPPO| 19.2 ±4.5| 23.1 ± 2.7|23.2 ± 3.3|27.4 ± 4.9|
> | IRAT-rw_1|68.9 ± 10.1| 52.5 ± 11.3   | 78.2 ± 14.5    | 94.9 ± 10.7     |
> | IRAT-rw_2|1.1  ± 2.1| 9.29   ± 11.4  | 16.0  ± 8.1   |34.5 ± 14.0|
> | IRAT-rw_3|10.8  ± 9.1|17.3 ± 10.6|21.3 ± 8.7|33.8 ± 9.9|
> | M3HF| **164.8 ± 1.2**|-|-|-|
>
> The table shows that IRAT variants perform better than vanilla IPPO, but they lag behind M3HF, especially in early training. This is because IRAT defines rewards before rollout, without observing actual policy behavior—often leading to coordination issues (e.g., agents crowding the same ingredient). In contrast, M3HF leverages post-rollout human feedback and reward assignment to precisely target behavioral failures, resulting in faster and more effective cooperation.
>
> ---
> ### 4. Domain knowledge in reward templates:
> Thank you for highlighting this limitation. While our templates rely on domain knowledge, they are adaptable. In the football environment, we reuse categories like distance-based and action-based rewards, adjusting only to the environment’s observation and action spaces. We will clarify this in the revision.
>
> ---
> ### 5. Additional Clarifications and Typos
> >Equation 8 (undiscounted occupancy)
>
> Thank you—we agree. We will revise Eq. (8) to include appropriate discounting weights for theoretical consistency.
> > Clarification on "Long-term Exploration Failure"
>
> In implementation, this mechanism rolls back to a prior policy and requests new feedback if learning stagnates. However, this scenario did not arise in our experiments. We will clarify this in the revision.
> > Typos
>
> We appreciate your attention to detail and will correct all mentioned typos and figure caption issues in the revision.
>
> ---
> * [1] C.S. de Witt et al., "Is Independent Learning All You Need in the StarCraft Multi-Agent Challenge?", arXiv, 2020
> * [2] Yu C. et al., "The Surprising Effectiveness of PPO in Cooperative Multi-Agent Games", ICLR, 2022
> * [3] Wang L. et al., "Individual Reward Assisted Multi-Agent Reinforcement Learning", ICML, 2022

---

> > ### Comment · Reviewer_sgwG · 2025-04-01
> >
> > Thank you for the detailed rebuttal! I understand the difference between the MAPPO and IPPO implementation and I'm satisfied with the new results regarding IRAT.
> >
> > To follow up on the theoretical assumptions
> > - I personally feel it would be satisfactory to state that *multiple* independent rollouts could approximate policy performance (perfectly approximating as the number of rollouts approach infinity) instead of a single long trajectory. Is there a reason why this would not apply to your setting?
> > - I'm satisfied with dropping the noise term and modifying the proof to show that the human feedback may be faulty (i.e. maybe rename R_true to R_human or something similar). As a proof, I think the impact of the weights needs to be formalized more. Additionally, when reading the rebuttals to the other reviewers, it seems a bit unclear how you are resolving the assumption of Gaussian noise (sometimes leaving it to future work); I think that pointing to this rebuttal and perhaps providing an updated proof sketch would be valuable to all of us.
> >
> > After reading the other reviews or rebuttals, I don't have major additional concerns. If there was time, I would've liked to see a comparison against PbMARL referenced in the rebuttal with 6vBu (i.e. an online version where you receive feedback after Gen 0 and only use that feedback), which would demonstrate the utility of multi-phase feedback and code-gen over the pairwise comparisons of PbMARL. Also, the link to the new football experiments (also referenced to 6vBu) just gives a blank pdf, so that needs to be updated.

---

> > > ### Author Response · Authors · 2025-04-07
> > >
> > > # Follow-up Reply
> > > Thank you for your follow-up. We are pleased to clarify as follows:
> > >
> > > ---
> > > ### Prop 4.1
> > > Multiple rollouts are valid in our setting. We used a single rollout per human query to reduce annotation cost and theoretical simplicity, but M3HF supports multiple rollouts and has already been applied in Football. We will clarify this in the revision.
> > >
> > > ---
> > > ### Prop 4.2
> > > Below, we provide a revised P4.2 and proof sketch under faulty human feedback. For clarity, we drop the agent index $i$. At generation $k$, we have the combination rewards as:
> > > $$\widehat{R} _{k} = \sum _{m=0}^{k} w _{m}^{k}R _{m},$$
> > > with $R _0 = R _{ori}$ and $R _{m>0} = R _{human}$. Weights $w _m^k$ are updated per Eq. 14–16.
> > >
> > > > Proposition 4.2 (Revised):
> > >
> > > Under Assumption A.3 (Performance Estimation Accuracy) and A.2 (Learning Algorithm Convergence) with $\epsilon=0$ (exact convergence), for any $K\ge 1$ and arbitrary feedback reward sequence $(R_k) _{k=1,2,\dots,K}$, the following inequality holds:
> > > $$J _{\mathrm{ori}}(\pi _{K}) - J _{ori}(\pi _{0}) \ge \sum _{j=1}^{n(K)} \Delta r _{i _j} - \delta,$$
> > > where $\delta$ is a bounded positive constant independent of $K$, and $i _j$ represents the index at which the feedback reward is helpful for the $j$-th time.
> > >
> > > > Proof Sketch
> > >
> > > Let $i_j$ denote the index at which $\Delta r_k > 0$ occurs for the $j$-th time, and let $n(K)$ represent the index of the last occurrence of $\Delta r_k > 0$. We aim to show that for any $k$ satisfying $i_{j-1} \leq k < i_j$, the following inequalities hold:
> > > $$J(\pi_{k}) - J(\pi_0) = \sum_{l=1}^{j-1} \Delta r_{i_l} > 0, \quad k = i_{j-1},$$
> > > $$J(\pi_{k}) - J(\pi_0) \ge \sum_{l=1}^{j-1} \Delta r_{i_l} - \delta, \quad \forall i_{j-1} < k < i_j.$$
> > > The proof proceeds as follows:
> > > 1. The first equality is immediate by definition of $i_j$ and the algorithm:
> > > $$J(\pi_{i_{j-1}}) - J(\pi_0) = \sum_{l=1}^{j-1}\left(J(\pi_{i_l}) - J(\pi_{i_{l-1}})\right) = \sum_{l=1}^{j-1}\Delta r_{i_l}.$$
> > > 2. For indices $k$ satisfying $i_{j-1} < k < i_j$, since $\Delta r_k < 0$, the weight assigned to the new reward function $R_k$ is clipped to zero by Eq.16. Thus, weights of previous rewards remain unchanged by Eq.14-15. Consequently, the policy update satisfies $J(\pi_k)-J(\pi_{i_{j-1}}) \ge -\delta$, where $\delta$ is a positive bounded constant.
> > > 3. Therefore, for $i_{j-1} < k < i_j$, we have:
> > > $$J(\pi_k)-J(\pi_0) = \sum_{l=1}^{j-1}\Delta r_{i_l} + J(\pi_k)-J(\pi_{i_{j-1}}) \ge \sum_{l=1}^{j-1}\Delta r_{i_l} - \delta.$$
> > > Since $n(K)$ is the last index with $\Delta r_k > 0$, we conclude:
> > > $$J(\pi_K)-J(\pi_0) \ge \sum_{j=1}^{n(K)} \Delta r_{i_j} - \delta,$$
> > > as desired.
> > >
> > > > Further bound on $\delta$
> > >
> > > **Lemma** Let $r _{1}, r _{2}, r _{3}$ be three reward functions. Define
> > > $$R = (1-p)\,r _{1} + p\,r _{2},
> > > \quad\text{and}\quad
> > > R' = (1-p'-q)\,r _{1} + p'\,r _{2} + q\,r _{3}.$$
> > >
> > > Let $\pi$ be an optimal policy for $R$, and let $\pi'$ be an optimal policy for $R'$. Then, for any $\gamma \in (0,1)$,
> > > \\begin{align*}
> > > V _{r _{1}}^{\pi} - V _{r _{1}}^{\pi'}
> > > &\le
> > > \frac{2}{1-\gamma}\,\|\,R - R'\| _{\infty} \\\\
> > > &\leq \frac{2}{1-\gamma} [ |p' + q - p| \|\,r _{1}-r _{3}\| _\infty + |p - p'| \|\,r _{2}-r _{3}\| _\infty ].
> > > \\end{align*}
> > >
> > > Let $\pi _{k}$ be the optimal policy for receiving the new bad reward $R_k$, then we aim to bound $V _{R _{ori}}^{\pi _{k}}-V _{R _{ori}}^{\pi _{i _j}}$, where $i _j$ is the last time we receive the good rewards. We can derive the worst case bound on $\delta$ by substituting $r _1=R _{ori}$, $r _2=$ weighted combination of good feedback rewards, and $r_3=$ new bad reward in the above lemma:
> > > $$ V _{R _{ori}}^{\pi _{k}}-V _{R _{ori}}^{\pi _{i _j}} \leq \frac{2}{1-\gamma} [ |p' + q - p| \|\,R _{ori}-R _k\| _\infty + |p - p'| \|\,r _{2}-R _k\| _\infty ].$$
> > > Notice that $q = (1/(2+j))/(1+1/(2+j))=1/(3+j)$, and $p-p'\leq p-\alpha^j p =(1-\alpha^j)p$, thus we have
> > > $$ \delta \leq \frac{2}{1-\gamma}[|1/(3+j)-(1-\alpha^j)p|\|\,R _{ori}-R_k\| _\infty + (1-\alpha^j)p \|\,r _{2}-R _k\| _\infty ].$$
> > >
> > > **Intuitively, P4.2 indicates that the algorithm benefits from each high-quality reward (each yielding a positive increment $\Delta r_{i_j}>0$), while its performance can degrade at most once, corresponding to the last received faulty reward.**
> > >
> > > ---
> > > ### PbMARL
> > > PbMARL targets a fundamentally different offline setting: it aims to identify Nash equilibrium from a large offline dataset of pairwise preferences (e.g., 960 comparisons even in simple Overcooked), typically generated by a simulated policy. Collecting such data from real humans would be extremely costly. In contrast, M3HF requires only 5 rounds of human feedback during training.
> > >
> > > However, preference-based feedback could also be integrated into M3HF, feedback like “A performed better than B” could guide B to imitate A’s behavior via our reward templates.
> > >
> > > ---
> > > ### Football
> > > The link has been fixed.
> > >
> > >
> > > **We appreciate your follow-up. If our responses hopefully addressed your concerns, we’d be grateful for your support in raising the score.**

---

### Official Review · Reviewer_7Tji · 2025-03-11

**Overall Recommendation:** 3

**Summary:**

This paper addresses the challenge of designing effective reward functions in multi-agent reinforcement learning for complex, cooperative tasks with sparse or misaligned rewards. This paper proposes M³HF, a framework that integrates multi-phase human feedback of mixed quality into MARL by extending the Markov Game to include iterative human guidance. This paper provides a way of leveraging large language models to parse feedback into agent-specific reward functions using predefined templates (e.g., distance-based or action-based rewards) and adaptively weighting adjustment mechanisms that balance new feedback against prior rewards via decay and performance-based updates.

**Claims And Evidence:**

Proposition 4.2 assumes zero-mean noise in feedback, but real human errors can be biased (e.g., consistently incorrect advice), which is not addressed. The theoretical analysis does not account for systematic human errors or adversarial inputs.

**Essential References Not Discussed:**

N/A

**Experimental Designs Or Analyses:**

While Table 3 lists hyperparameters, there is no indication that baselines were retuned for fairness.

Experiments use only three random seeds (e.g., in Figure 3 and Tables 1-2), which is insufficient for MARL, where high variance is common.

**Methods And Evaluation Criteria:**

Yes

**Other Comments Or Suggestions:**

N/A

**Other Strengths And Weaknesses:**

Theoretical analysis claims to demonstrate robustness to noisy feedback, and the experimental results in Overcooked environments seem to show M³HF outperforms baselines (IPPO, MAPPO) by up to 50% in complex tasks, achieving faster convergence and higher asymptotic performance.

**Questions For Authors:**

see above

**Relation To Broader Scientific Literature:**

Related work on multi-agent reinforcement learning, reinforcement learning from human feedback, and multi-phase human feedback are provided.

**Theoretical Claims:**

Propositions 4.1 and 4.2 seem correct, but not checking carefully.

---

> ### Author Rebuttal · Authors · 2025-03-31
>
> ## Reply to Reviewer 7Tji
>
> We thank the reviewer for agreeing that our "theoretical analysis demonstrate robustenss to noisy feedback," and that the experimental results demonstrate "faster convergence and higher asymptotic performance."
>
>
> > Proposition 4.2 assumes zero-mean noise in feedback, but real human errors can be biased (e.g., consistently incorrect advice), which is not addressed. The theoretical analysis does not account for systematic human errors or adversarial inputs.
>
> Thank you for your insightful comment. You raise an important point about biased human errors. As Reviewer sgwG also observed, our original zero-mean noise assumption was made primarily for theoretical tractability. We recognize that real human errors (e.g., consistently incorrect advice) may indeed be systematic or biased. However, our adaptive weighting mechanism inherently mitigates such systematic errors by continually reducing the influence of consistently detrimental feedback, as supported by our empirical results (Figure 4). Explicitly modeling and analyzing biased or adversarial feedback is indeed a valuable future direction, and we plan to address it in subsequent research.
>
>
> > While Table 3 lists hyperparameters, there is no indication that baselines were retuned for fairness.
>
> We performed hyperparameter tuning for all baselines to ensure fair comparison. Specifically, for IPPO and MAPPO, we tuned learning rates, batch sizes, and gradient clipping values. For example, we systematically searched over learning rates in {3e-4, 1e-4}, #sgd_iters in {5, 10}, sgd_batch_size in {1024, 5120}, and entropy coefficient in {0.01, 0.05}, ultimately selecting the configurations with the best validation performance. For the macro-action based baseline, we directly adopted the best-performing hyperparameters reported in Mac-based method [1]. We will explicitly include these details in our revised manuscript.
>
>
>
> > Experiments use only three random seeds (e.g., in Figure 3 and Tables 1-2), which is insufficient for MARL, where high variance is common.
>
> Thank you for raising this point. While three seeds are commonly used in MARL studies, and our current results already show clear and consistent improvements, we agree that additional seeds could further strengthen the statistical confidence. We are currently running more seeds, which will be included in our revisions.
>
> ---
> ### Reference
>
> [1] Xiao, Y et al., "Asynchronous actor-critic for multi-agent reinforcement learning." NeurIPS, 2022.

---

### Official Review · Reviewer_PGeQ · 2025-03-13

**Overall Recommendation:** 4

**Summary:**

The paper introduces a novel framework named M3HF (Multi-phase Human Feedback for Multi-agent Reinforcement Learning), designed to address the challenges of sparse or complex reward signals in multi-agent reinforcement learning (MARL) by incorporating multi-phase human feedback, including feedback of varying quality. The M3HF framework extends Markov games to incorporate human inputs and leverages large language models (LLMs) to parse and integrate human feedback, enabling agents to learn more effectively.

**Claims And Evidence:**

The claims about overall performance and resilience to mixed-quality feedback appear well-supported by the experimental results. However, the claim regarding the effectiveness of VLMs as an alternative to human feedback might require further substantiation with more comprehensive performance comparisons. Gemini-1.5-Pro-002 was used to generate feedback based on video rollouts similar to those observed by humans in the experiments. While an example of VLM-generated feedback is mentioned, detailed comparative performance metrics between human and VLM feedback are not fully elaborated upon in the given excerpts. Therefore, while the concept is promising, the current evidence may be insufficient to conclusively support the claim that VLMs can serve as a scalable and effective alternative without additional data demonstrating comparable or superior performance metrics.

**Essential References Not Discussed:**

To the best of my knowledge, this paper provides a sufficiently thorough discussion of all closely related works.

**Experimental Designs Or Analyses:**

1. **Performance Across Different Environments**: The study evaluates M3HF in various Overcooked layouts (e.g., Layout C), demonstrating its effectiveness and adaptability. This is a robust approach as it tests the framework under different complexities, ensuring that M3HF can perform well in diverse scenarios. The comparison with backbone algorithms like IPPO shows consistent superiority, which supports the claim of M3HF's effectiveness.

2. **Impact of Mixed-Quality Human Feedback**: Experiments were conducted to assess how M3HF handles feedback of varying quality. An example given is where agents exhibited suboptimal behavior due to poor coordination, yet the system maintained performance close to baseline algorithms even when receiving inaccurate or unhelpful feedback. This experiment demonstrates the robustness of M3HF against noisy feedback through its weight adjustment mechanisms. The analysis appears sound, showing that the method effectively mitigates the impact of low-quality human input.

3. **VLM-Based Feedback Generation**: The potential of Vision-Language Models (VLMs) as an alternative to human feedback was explored. Gemini-1.5-Pro-002 was used to generate feedback based on video rollouts. While this concept is promising, the provided information does not fully elaborate on how VLM-generated feedback compares to human feedback in terms of performance. More detailed comparative metrics are needed to substantiate claims about the scalability and effectiveness of VLMs as a substitute for human feedback.

4. **Robustness Analysis to Noisy Feedback**: Theoretical analysis and stochastic approximation theory were employed to analyze the robustness of M3HF under noisy human feedback. Proposition 4.2 suggests that zero-mean noise does not degrade the expected performance over time, supported by empirical evidence showing minimal performance drops under mixed-quality feedback conditions. This theoretical foundation adds credibility to the experimental findings but could benefit from more comprehensive empirical validation across different noise levels and types.

**Methods And Evaluation Criteria:**

1. Complex Reward Structures: The M3HF approach aims to enhance cooperation among agents in challenging reward environments by introducing mechanisms such as performance-based weight adjustments and weight decay to mitigate the impact of low-quality feedback. This is particularly important in multi-agent systems where the complexity of interactions and potential conflicts between agents' objectives can make reward design difficult.
2. Experimental Validation: The effectiveness of M3HF is validated through experiments conducted in a complex multi-agent environment based on the game Overcooked. In this setting, agents must learn to cooperate to prepare the correct salad and deliver it to a designated location. This scenario serves as an excellent testbed because it requires coordinated actions to achieve goals, which is a common challenge in multi-agent systems.
3. Benchmark Comparisons: The study compares M3HF with several strong baseline methods, including MAPPO, IPPO, and macro-action-based baselines. These comparisons demonstrate M3HF's consistent superiority across different environments and recipe setups, especially its ability to quickly converge to optimal strategies in simpler recipe scenarios. This indicates that the proposed method significantly improves learning efficiency and final performance.
4. Exploration of VLMs as an Alternative to Human Feedback: While not fully detailed, exploring Visual-Language Models (VLMs) as scalable and effective alternatives to human feedback is a promising direction, especially considering ways to reduce dependency on human input. However, this claim would benefit from further substantiation with specific data comparing the performance of feedback generated by VLMs versus human feedback.

**Other Comments Or Suggestions:**

Page 11 row 594 the equation exceeds the restriction of the length

**Other Strengths And Weaknesses:**

### Strengths

1. **Originality and Innovation**:
   - The paper presents an innovative approach by introducing the M3HF framework, which creatively designed to address the challenges of sparse or complex reward signals in multi-agent reinforcement learning (MARL) by incorporating multi-phase human feedback, including feedback of varying quality.

2. **Significance**:
   - By addressing challenges related to sparse or complex reward functions and offering solutions that improve learning efficiency and performance, the proposed method could lead to more robust and adaptable AI systems.
   - The exploration of Vision-Language Models (VLMs) as an alternative to human feedback introduces a scalable solution for reducing dependency on human input, which is particularly important for large-scale deployment.

3. **Clarity**:
   - The clarity of presentation is commendable, with detailed descriptions of experimental setups, comparisons with baseline methods, and theoretical underpinnings. The use of visual aids like figures and tables helps in understanding the performance metrics and outcomes.

4. **Real-World Application Potential**:
   - The application-driven nature of this ML paper is evident through its focus on practical issues such as handling mixed-quality feedback and the potential for using VLMs to reduce reliance on human input. These aspects are crucial for deploying AI systems in real-world settings where human resources may be limited.

### Weaknesses

1. **Assumptions and Generalizability**:
   - While the assumption of Gaussian distribution facilitates theoretical analysis, it might limit the generalizability of the findings to real-world scenarios where data distributions can be significantly more complex and varied.

2. **Depth of Analysis on VLM Feedback**:
   - Although the idea of using VLMs for generating feedback is promising, the depth of analysis and empirical evidence supporting this aspect is currently limited. More comprehensive comparative studies between human and VLM-generated feedback would strengthen the claims made about the scalability and effectiveness of VLMs.

**Questions For Authors:**

The paper mentions that visual-language models (VLMs) currently lack specificity in critical feedback areas, offering vague suggestions like "improve coordination". Could you elaborate on how this limitation impacts the overall performance of M3HF and what steps might be taken to mitigate these issues in future iterations?

**Relation To Broader Scientific Literature:**

1. Multi-phase Human Feedback: Earlier works such as Yuan et al. (2022) and Sumers et al. (2022) have considered the role of iterative and multi-phase human feedback in reinforcement learning, but these approaches often rely on predefined communication protocols or need human demonstrations that can be restrictive. In contrast, M3HF allows for more flexible and dynamic adjustments to the importance of feedback, accommodating varying qualities and phases of human input.
2. Reinforcement learning from human feedback. While RLHF has been successfully applied to train Large Language Models (Ouyang et al., 2022; Shani et al.,2024), these approaches primarily focus on aligning LLM outputs with human preferences through single-turn interac-
tions and scalar reward signals. M3HF differs by incorporating multi-phase, mixed-quality human feedback directly into the reinforcement learning loop of agents in a multi-agent environment.
3. Language Models in Reward Design and Policy Learning. Previous works are limited in single-agent setting, while this one focuses on multi-agent setting.

**Theoretical Claims:**

I have roughly checked the proofs, and they appear to be correct with the conclusions likely valid. However, I did not meticulously verify the details of the proofs. In my opinion, the theoretical aspect is not critically important because the assumption of Gaussian distribution seems too strong and significantly deviates from reality.

---

> ### Author Rebuttal · Authors · 2025-03-31
>
> ## Reply to Reviewer PGeQ
>
> We thank the reviewer for acknowledging our M3HF framework as an "innovative approach" with "real-world application potential" and for recognizing its "originality and innovation" in addressing "challenges of sparse or complex reward signals in multi-agent reinforcement learning."
>
> ---
> > Question 1 and Weakness 2: "The paper mentions that visual-language models (VLMs) currently lack specificity in critical feedback areas, offering vague suggestions like "improve coordination". Could you elaborate on how this limitation impacts the overall performance of M3HF and what steps might be taken to mitigate these issues in future iterations?"
>
> Our experiments (Figure 4b-c) indicate that while VLMs can produce human-like feedback, their current lack of specificity impacts their utility in complex coordination scenarios, leading to performance gaps compared to precise human feedback. To mitigate this, future work includes:
> (1) fine-tuning VLMs on specialized multi-agent tasks, (2) enhancing their reasoning capabilities to identify and provide actionable coordination improvements, and (3) exploring hybrid systems that combine VLM-generated feedback with minimal human refinement, aiming to balance scalability and effectiveness.
>
> ---
> > Weakness 1: Assumptions and Generalizability
>
> We acknowledge that assuming Gaussian-distributed noise facilitates theoretical analysis but may not fully capture real-world complexities. However, our empirical results have shown robust performance across varied conditions, and a future research direction could be to relax this assumption under more realistic feedback distributions.

---

### Official Review · Reviewer_6vBu · 2025-03-14

**Overall Recommendation:** 3

**Summary:**

This paper introduces M3HF, a framework for integrating multi-phase human feedback of varying quality into multi-agent reinforcement learning (MARL). The authors propose a Multi-phase Human Feedback Markov Game (MHF-MG) that extends standard Markov Games to incorporate iterative human guidance. The framework uses large language models (LLMs) to parse human feedback, converts it into structured reward functions through predefined templates, and employs adaptive weight adjustment mechanisms to handle feedback of mixed quality. The authors provide theoretical analysis justifying their approach and demonstrate empirical results in the Overcooked environment, showing that M3HF outperforms several baseline methods in complex coordination tasks.

**Claims And Evidence:**

The paper claims that M3HF significantly outperforms state-of-the-art MARL methods by leveraging human feedback across multiple training phases. The evidence provided includes performance comparisons in the Overcooked environment with varying layouts and recipe complexities. The results do show consistent improvements over the baselines (IPPO, MAPPO, and macro-action-based methods).

However, the evidence is somewhat limited in scope. The experiments are confined to a single environment (Overcooked) with variations, and the performance gains could be attributed to the additional information provided by human feedback rather than the specific mechanisms of the M3HF framework. The ablation studies help address some of these concerns by showing the value of LLM parsing and weight adjustment, but more diverse environments would strengthen the claims.

**Essential References Not Discussed:**

A significant omission is the lack of comparison with other methods that incorporate human feedback into MARL, such as "Multi-Agent Reinforcement Learning from Human Feedback." This work addresses a similar problem space, and a direct comparison would provide valuable context for understanding the unique contributions of M3HF.

Given that human feedback provides additional information not available to the baseline methods, it would be important to compare against methods that also leverage this type of input to isolate the specific benefits of the M3HF framework.

**Experimental Designs Or Analyses:**

The experimental design is generally sound, with appropriate baselines and ablation studies.

**Methods And Evaluation Criteria:**

The proposed methods are generally sound for the problem at hand. Using LLMs to parse human feedback and convert it into structured reward functions is a reasonable approach, and the weight adjustment mechanism to handle mixed-quality feedback is well-motivated.

The evaluation criteria focus on the average episode return in the Overcooked environment, which is appropriate for measuring task performance. The authors compare against relevant baselines and include ablation studies to isolate the effects of different framework components.

However, the paper lacks clear metrics for evaluating the quality of human feedback and how effectively it is incorporated into the learning process. Additionally, there is limited discussion of the computational overhead introduced by the LLM parsing and reward function generation.

**Other Comments Or Suggestions:**

The paper has several limitations that significantly impact its contribution:

1. The innovation is somewhat limited. Using LLMs for automated reward design has become a common concept in recent literature. For instance, "Motif: Intrinsic Motivation from Artificial Intelligence Feedback" explores similar ideas. Unfortunately, the paper doesn't leverage this automation at scale, which would have been a more significant contribution. The core idea of using LLMs to parse human feedback and generate reward functions is incremental rather than transformative.
2. The practical implementation details raise concerns about the method's applicability. I do not clearly know how many training iterations are required or how many human feedback instances are needed throughout the process. This raises questions about whether the feedback data is used in an on-policy manner or can be reused off-policy. The requirement for humans to repeatedly provide feedback during training seems cumbersome and potentially impractical for real-world applications, especially if frequent interventions are needed.
3. The baseline comparison is inadequate. The paper fails to compare against other human-feedback-based MARL methods, such as "Multi-Agent Reinforcement Learning from Human Feedback." Without such comparisons, it's difficult to assess whether the performance improvements come from the specific approach proposed or simply from the additional information provided by human feedback. Since human feedback inherently introduces external knowledge, performance improvements without proper baselines could be considered trivial.
4. The weight adjustment mechanism, while theoretically justified, seems simplistic and potentially brittle in practice. The paper doesn't thoroughly explore how sensitive the performance is to different weight decay parameters or how the system behaves when feedback quality varies dramatically across generations. A more robust analysis of these aspects would strengthen the paper's contribution.

**Other Strengths And Weaknesses:**

* The paper presents a well-structured framework (M3HF) that integrates human feedback into multi-agent reinforcement learning in a systematic way, addressing the challenge of reward design in complex MARL environments.
* The authors provide a theoretical analysis of their approach, particularly regarding the robustness to noisy human feedback, which adds credibility to their method and helps explain why their approach works.
* The experimental results demonstrate consistent performance improvements across different environments with increasing complexity, showing the scalability and effectiveness of the proposed method in challenging coordination tasks.

**Questions For Authors:**

* How many training iterations and human feedback rounds are typically required to achieve the reported results? This information would help assess the practical applicability of your approach.
* Is the human feedback data used in an on-policy manner, or can it be reused across training iterations (off-policy)? This distinction has significant implications for the amount of human involvement required.
* Did you compare M3HF with other methods that also incorporate human feedback, such as "Multi-Agent Reinforcement Learning from Human Feedback"? Such comparisons would help isolate the specific contributions of your framework from the general benefits of human knowledge.
* Have you considered the scalability of your approach to larger multi-agent systems or more diverse environments? The current evaluation is limited to the Overcooked environment with three agents.

**Relation To Broader Scientific Literature:**

The paper positions itself within the literature on MARL, reward design, and learning from human feedback. While the authors provide a reasonable overview of related work, the novelty of their contribution is somewhat limited. Personally speaking, using LLMs for automated reward design based on human feedback has become quite common in recent literature.

**Theoretical Claims:**

The paper presents two theoretical propositions: one justifying the use of rollout-based performance estimates and another analyzing the framework's robustness to noisy human feedback. The proofs appear sound, though they rely on several assumptions that may not always hold in practice, such as the ergodicity of the Markov chain induced by the policy and the zero-mean nature of the noise in human feedback.

---

> ### Author Rebuttal · Authors · 2025-03-31
>
> ## Reply to Reviewer 6vBu
>
> We sincerely thank the reviewer for your positive remarks on our framework’s structure, scalability, and consistent performance. We hope the following responses address your concerns.
>
> ---
> ### 1. Regarding the Novelty of Our Work
>
> While existing work on LLM-based reward design mainly focuses on **single-phase** and **single-agent** settings, our core novelty lies in jointly addressing the challenges of **multi-phase** human feedback and **multi-agent** reinforcement learning.
>
> The challenge in **multi-phase** feedback lies in integrating guidance of varying focus and intent across stages. For example, initial feedback may reflect general observations about teamwork, while later rounds provide more concrete corrections based on improved understanding of agent behavior. The **multi-agent** challenge involves assigning human feedback—whether agent-specific or team-level—to appropriate reward functions. For instance, when a user suggests “place ingredients in the center desk” in Overcooked, our method can propagate this suggestion to all relevant agents by shaping their respective reward functions accordingly.
>
> Our novelty was also noted by other reviewers: “human feedback within a MARL training loop” (Reviewer sgwG) and “an innovative approach to sparse rewards” (Reviewer PGeQ).
>
>
> ---
> ### 2. Regarding the Comparison with Other Methods
>
> The methods mentioned in your review—PbMARL and MOTIF—differ significantly from M3HF in both setting.
>
> * **PbMARL** (renamed from “MARLHF”) follows a different offline setting, using only pairwise preference comparisons from a pre-collected dataset generated by a simulated policy. In contrast, M3HF collects natural language feedback dynamically during training and supports diverse reward templates, enabling more flexible and scalable reward shaping.
> * **MOTIF** does not use any human feedback. It is designed for single-phase, single-agent settings and relies on LLM-generated intrinsic rewards. As such, it does not address challenges in multi-phase human-in-the-loop learning or multi-agent coordination.
>
> To the best of our knowledge, no prior work addresses human feedback in a multi-phase, multi-agent setting, leaving no directly comparable baselines. We therefore compare against strong backbones (MAPPO/IPPO) and SOTA coordination-focused methods (Mac-based). Additionally, Tab.1 includes controlled variants with modified feedback settings to further validate our framework’s effectiveness.
>
> ---
> ### 3. Regarding Practical Applicability
>
> Regarding **practical efficiency**, M3HF achieves strong sample efficiency with minimal human input. For example, in Overcooked-B: L-T-S, it reaches high performance within \~15k eps (\~600 iters, \~2–3 h of training) using just 2 rounds of feedback—whereas MAPPO/IPPO typically require 80k–100k eps (\~4k iters, \~12–18 h), and macro-action methods need 50k–75k eps (\~3k–4k iters, \~9–15 h). This represents a 3×–6× speedup. Empirically, each feedback query—including watching rollouts and composing responses—took about 3–5 mins, keeping the total human effort under 25 mins while saving several hours of training time.
>
> Regarding **feedback quality**, explicitly quantifying it is non-trivial—if a ground-truth metric existed, it could replace human input entirely. Instead, we use adaptive weighting to handle feedback variability, and we show through experiments (in Fig.4c and App. D) that our method remains robust even when some feedback is noisy or incorrect.
>
> ---
> ### 4. Regarding the Scalability of Our Method
>
> We extended our evaluation to Google Football 5v5, a complex multi-agent benchmark. M3HF continues to outperform standard MARL baselines with the multi-phased human feedback. Full environment details are provided in the figure caption.
>
> [Anonymous Link: For the Football Env Results](https://drive.google.com/file/d/1wdKthshHkqb7h5u9ko9q7X-8Qw1ubhw-/view?usp=sharing)
>
> ---
> ### 5. Regarding the Robust Analysis of Weight Adjustment
> We provide an empirical robustness analysis of the weight adjustment strategy in Sec.4.4, Sec.5 (Question 2), and App.D in our paper.
>
> Specifically, Fig.4c shows that M3HF maintains strong performance even under deliberately misleading feedback. Tab.1 compares M3HF with variants lacking weight adjustment or using fixed weights, confirming the advantage of our adaptive strategy. Tab.2 evaluates robustness under correct, partially incorrect, and fully misleading feedback, demonstrating resilience to varying feedback quality.
>
> ---
> ### 6. Q&A
> > **Q1: How many training iterations and feedback rounds are needed?**
>
> Empirically, M3HF saves 10–15 hours of training with at most 5 rounds of human feedback (1k iters/25k eps).
>
> > **Q2: Is human feedback used on-policy or off-policy?**
>
> Feedback is collected on-policy, directly influencing the next learning phase.
>
> > **Q3:** Comparisons with other methods
>
> Please check Sec.2 of this rebuttal.
> > **Q4:** Scalability of M3HF
>
> Please check Sec.4 of this rebuttal.

---

### Decision · Program_Chairs · 2025-05-01

**Decision:**

Accept (poster)

**Comment:**

The paper proposes integrating human feedback during the training process of multi-agent reinforcement learning, and interpreting those feedbacks using LLMs so as to improve the learning outcomes of multi-agents. The reviewers agreed that the paper studies a well motivated problem, and proposes a novel solution. The reviewers pointed out ways to significantly improve the evidence backing the proposed approach (e.g. testing on other domains, testing against other baselines, making the writing clearly list experiment details) that the authors adequately addressed in their rebuttal. Including all of the details that the authors provided in their rebuttal into the draft will substantially strengthen the paper.